# Valid Conformal Prediction for Dynamic GNNs

**Ed Davis**[1]    **Ian Gallagher**[2]    **Daniel John Lawson**[1]    **Patrick Rubin-Delanchy**[3]
[1]University of Bristol, U.K.    [2]The University of Melbourne, Australia
[3]School of Mathematics and Maxwell Institute for Mathematical Sciences,
University of Edinburgh, U.K.
{edward.davis, dan.lawson}@bristol.ac.uk
ian.gallagher@unimelb.edu.au
prd@ed.ac.uk

## Abstract

Dynamic graphs provide a flexible data abstraction for modelling many sorts of real-world systems, such as transport, trade, and social networks. Graph neural networks (GNNs) are powerful tools allowing for different kinds of prediction and inference on these systems, but getting a handle on uncertainty, especially in dynamic settings, is a challenging problem.

In this work we propose to use a dynamic graph representation known in the tensor literature as the unfolding, to achieve valid prediction sets via conformal prediction. This representation, a simple graph, can be input to any standard GNN and does not require any modification to existing GNN architectures or conformal prediction routines.

One of our key contributions is a careful mathematical consideration of the different inference scenarios which can arise in a dynamic graph modelling context. For a range of practically relevant cases, we obtain valid prediction sets with almost no assumptions, even dispensing with exchangeability. In a more challenging scenario, which we call the semi-inductive regime, we achieve valid prediction under stronger assumptions, akin to stationarity.

We provide real data examples demonstrating validity, showing improved accuracy over baselines, and sign-posting different failure modes which can occur when those assumptions are violated.

## 1    Introduction

Graph neural networks (GNNs) have seen success in a wide variety of application domains including prediction of protein structure (Jumper et al., 2021) and molecular properties (Duvenaud et al., 2015; Gilmer et al., 2017), drug discovery (Stokes et al., 2020), computer vision (Sarlin et al., 2020), natural language processing (Peng et al., 2018; Wu et al., 2023), recommendation systems (Berg et al., 2017; Wu et al., 2022; Borisyuk et al., 2024), estimating time of arrival (ETA) in services like Google Maps (Derrow-Pinion et al., 2021), and advancing mathematics (Davies et al., 2021). They often top leaderboards in benchmarks relating to machine-learning on graphs (Ope), for a range of tasks such as node, edge, or graph property prediction. Useful resources for getting started with GNNs include the introductions (Hamilton, 2020; Sanchez-Lengeling et al., 2021) and the Pytorch Geometric library (Pyt).

Conformal prediction (CP), advanced by Vovk and co-authors (Vovk et al., 2005), and later expanded on by various researchers in the Statistics community (Shafer and Vovk, 2008; Lei et al., 2013; Lei and Wasserman, 2014; Lei et al., 2018; Foygel Barber et al., 2021; Tibshirani et al., 2019; Barber et al., 2023; Gibbs et al., 2023; Romano et al., 2020), is an increasingly popular paradigm for constructing provably valid prediction sets from a 'black-box' algorithm with minimal assumptions and at low cost. Several recent papers have brought this idea to quantifying uncertainty in GNNs (Huang et al., 2024b; Clarkson, 2023; Zargarbashi and Bojchevski, 2023).

There are many applications of significant societal importance, e.g. cyber-security (Bilot et al., 2023; He et al., 2022; Bowman and Huang, 2021), human-trafficking prevention (Szekely et al., 2015; Nair et al., 2024), social media & misinformation (Phan et al., 2023), contact-tracing (Tan et al., 2022), patient care (Liu et al., 2020; Boll et al., 2024) where we would like to be able to use GNNs, with uncertainty quantification, on dynamic graphs.

The added time dimension in dynamic graphs causes serious issues with existing GNN + conformal methodology, broadly relating to de-alignment between embeddings across time points (see Figure 1), resulting in large or invalid prediction sets in even simple and stable dynamic regimes. The unsolved problem we address, in a nutshell, is guaranteeing that two nodes behaving in the same way at distinct points in time are embedded in an exchangeable way, regardless of global network dynamics.

Inspired by a concurrent line of statistical research on spectral embedding and tensor decomposition (Zhang and Xia, 2018; Cape et al., 2019; Abbe et al., 2020; Rubin-Delanchy et al., 2022; Jones and Rubin-Delanchy, 2020; Gallagher et al., 2021; Agterberg and Zhang, 2022), we propose to use one of the *unfoldings* (De Lathauwer et al., 2000) of the tensor representation of the dynamic graph as input to a standard GNN, and demonstrate validity in both transductive and semi-inductive regimes.

**Related work.** The use of (tensor) unfoldings in GNNs is new. At present, to our knowledge, the only paper studying conformal inference on dynamic graphs is (Zargarbashi and Bojchevski, 2023), but the paper considers a strictly growing graph, which is very different from our meaning, in which a 'dynamic graph' has edges which appear and disappear in complex and random ways, as in all of our examples and in the applications alluded to above. The motivation for our work derives from earlier contributions studying CP on static graphs using GNNs (Huang et al., 2024b; Clarkson, 2023; Zargarbashi and Bojchevski, 2023) but our theory is distinct from those contributions, with arguments closer to (Barber et al., 2023). In the dynamic graph literature, unfoldings have exclusively been proposed for unsupervised settings (Jones and Rubin-Delanchy, 2020; Gallagher et al., 2021; Agterberg and Zhang, 2022; Davis et al., 2023; Wang et al., 2023), particularly spectral embedding, and have never been applied to conformal inference.

Our contribution should be viewed as a novel *interface* between CP and GNNs, rather than an improvement of either. This interface allows for a modular system design in dynamic graph inference in which both the GNN and CP modules can be chosen and optimised independently. In view of the rapid development of both fields, we have opted against fine-tuning to allow more transparent, stable, and reproducible performance assessments using recognised and standard baselines for GNNs and CP: Graph Convolutional Networks (GCN) (Kipf and Welling, 2016) and Graph Attention Networks (GAT) (Veličković et al., 2017) for GNNs and APS (Romano et al., 2020) for CP. Our theory applies to the *representation* of the dynamic graph, not the choice of GNN or CP procedures.

## 2 THEORY & METHODS

**Problem setup.** In this paper, a dynamic graph $G$ is a sequence of $T$ graphs over a globally defined nodeset $[n] = \{1, \ldots, n\}$, represented by adjacency matrices $\mathbf{A}^{(1)}, \ldots, \mathbf{A}^{(T)}$. Technically, all we require is that the $\mathbf{A}^{(t)}$ have the same number of rows, $p$, allowing for directed, bipartite and weighted graphs. However, the reader may find it easier to assume the $\mathbf{A}^{(t)}$ are $n \times n$ binary symmetric matrices corresponding to undirected graphs, in which case $p = n$.

Along with the dynamic graph $G$, we are given labels for certain nodes at certain points in time, which could describe a state or behaviour. Our task is to predict the label for a different node/time pair, whose label is hidden or missing. Our prediction must take the form of a set which we can guarantee contains the true label with some pre-specified probability $1 - \alpha$ (e.g. 95%).

We track the labelled and unlabelled node/time pairs in this problem using a fixed sequence $\nu$ where for each $(i, t) \in \nu$ we assume existence of

1. a class label $Y_i^{(t)} \in \mathcal{Y}$, which may not be observed, in some discrete set of cardinality $d$;

2. a set of attributes, which is observed, and represented by a feature vector $X_i^{(t)} \in \mathbb{R}^c$;

3. a corresponding column in $\mathbf{A}^{(t)}$, denoted $\mathbf{A}_{\cdot i}^{(t)}$, with at least one non-zero entry.

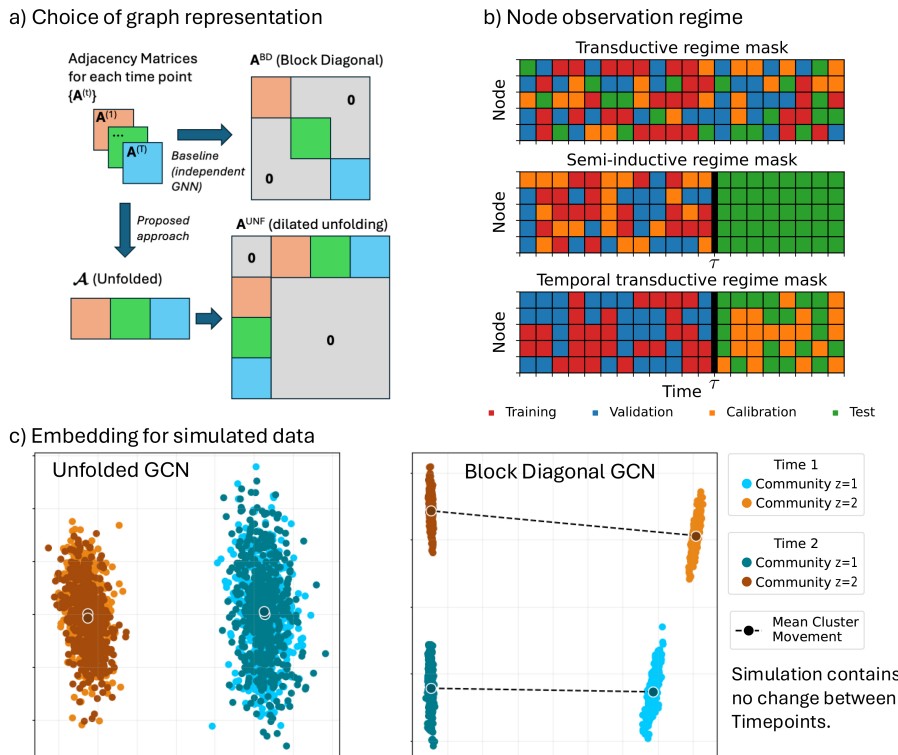

Figure 1: Contribution overview. a) This paper is about the representation of the collection of adjacency matrix snapshots. The baseline (current practice) approach treats these as independent and can be viewed as padding a 'block-diagonal' matrix with zeroes. Unfolding instead column concatenates which links nodes to themselves over time. Dilation results in a square symmetric matrix. b) Which data are available at training time affects performance; we report results for transductive (all time-points are exchangeable in terms of test/train split), semi-inductive (a future period is reserved for testing), or temporal transductive (a future period is reserved for testing and calibration). c) Simulation of an i.i.d. stochastic block model showing the embedding after applying PCA. The models were trained with transductive masks. Block diagonal GCN appears to encode a significant change over time despite there being none. The embedding from UGCN is exchangeable over time, as would be expected.

For a pair $w = (i, t) \in \nu$, we will use the notation $X_w := X_i^{(t)}, Y_w := Y_i^{(t)}, \mathbf{A}_w := \mathbf{A}_i^{(t)}$.

The data are split into four sets: $\nu^{(\text{training})}$ to learn the GNN parameters; $\nu^{(\text{validation})}$ for out-of-sample performance evaluation of the GNN; $\nu^{(\text{calibration})}$ to train the conformal model, and $\nu^{(\text{test})}$ is the target for conformal inference.

Clearly, the theory and methodology of this paper are applicable to multiple test nodes, but for simplicity in this section we assume $\nu^{(\text{test})}$ comprises only one node.

From these objects we define three versions of the inference problem.

*Transductive regime.* The test label is missing completely at random allowing training, validation, and calibration at any time (Figure 1b, top).

*Temporal transductive regime.* Training and validation are undertaken up to some time $\tau$. The test label is missing completely at random after time $\tau$, allowing calibration from this interval (Figure 1b, bottom).

*Semi-inductive regime.* Training and validation is undertaken up to some time $\tau$. The test label occurs after time $\tau$, corresponding to a fixed node $i$, preventing calibration from this interval (Figure 1b, middle).

In either of the transductive regimes, a valid confidence set can be constructed with no further assumptions (Algorithm 1). We believe this could be exciting because we view the temporal transductive regime, in particular, as an extremely common scenario. The semi-inductive regime is more challenging, and reflects a scenario where we have only historical labels. Here, we will need to invoke further assumptions such as symmetry and exchangeability, familiar in the conformal prediction literature.

Formally, for some $m < |\nu| - 1$, we reorder the dataset $\nu$ so that the first $m$ node/time pairs correspond to the calibration and test sets, while the remaining pairs correspond to the training and validation sets. For both transductive regimes, the missing label is missing at random among the first $m$ node/time pairs of $\nu$; for the semi-inductive regime, the missing label corresponds to a fixed node/time pair in this set. Let $\nu^{(\text{test})} := \{w_{\text{test}}\}$ denote the missing node/time pair, $\nu^{(\text{calibration})} := \{w_\ell, \ell \leq m\} \setminus \nu^{(\text{test})}$, and $\nu^{(\text{training})} \cup \nu^{(\text{validation})} := \{w_\ell, \ell > m\}$.

**Proposed approach.** We will repurpose a standard GNN, designed for static graphs, to produce dynamic graph embeddings with desirable exchangeability properties. Let $\mathcal{G}$ denote a GNN taking as input an adjacency matrix corresponding to an undirected graph on $n$ nodes, along with node attributes and node labels, and returning an *embedding* $\hat{\mathbf{V}} \in \mathbb{R}^{n \times d}$. Typically, the $i$th row of $\hat{\mathbf{V}}$ predicts the label of node $i$ through

$$\hat{\mathbb{P}}(\text{node } i \text{ has label } k) = \left[\text{softmax}(\hat{\mathbf{V}}_{i1}, \ldots, \hat{\mathbf{V}}_{id})\right]_k.$$

To represent the adjacency matrix we apply dilated unfolding (Davis et al., 2023) representing the dynamic graph $G$ in the form

$$\mathbf{A}^{\text{UNF}} := \begin{bmatrix} \mathbf{0} & \mathcal{A} \\ \mathcal{A}^\top & \mathbf{0} \end{bmatrix} \in \mathbb{R}^{(p+nT) \times (p+nT)}$$

where $\mathcal{A} = (\mathbf{A}^{(1)}, \ldots, \mathbf{A}^{(T)}) \in \mathbb{R}^{p \times nT}$ (column-concatenation) containing $p$ global (i.e. not t dependent) nodes and $nT$ temporal nodes (each a node/time pair).

The GNN takes as input a matrix $X^{\text{UNF}} \in \mathbb{R}^{(p+nT) \times c}$ containing $c$-dimensional attributes for all nodes in the dilated unfolding. For networks without attributes this matrix is the $(p+nT) \times (p+nT)$ identity matrix. The GNN also requires training and validation labels $Y_w, w \in \nu^{(\text{training})} \cup \nu^{(\text{validation})}$, where global nodes are not assigned a label. The resulting output splits into embeddings

$$\begin{bmatrix} \hat{\mathbf{U}}^{\text{UNF}} \\ \hat{\mathbf{V}}^{\text{UNF}} \end{bmatrix} := \mathcal{G}\left(\mathbf{A}^{\text{UNF}}, X^{\text{UNF}}, \{Y_w : w \in \nu^{(\text{training})} \cup \nu^{(\text{validation})}\}\right),$$

where $\hat{\mathbf{U}}^{\text{UNF}} \in \mathbb{R}^{p \times d}$ contains global representations and the embedding $\hat{\mathbf{V}}^{\text{UNF}} \in \mathbb{R}^{nT \times d}$ contains a representation of each node/time pair, of principal interest here.

Finally, we shall require some non-conformity score function $r : \mathbb{R}^d \times \mathcal{Y} \to \mathbb{R}$, with the convention that $r(\hat{y}, y)$ is large when $\hat{y}$ is a poor prediction of $y$. In practice, we make use of the adaptive non-conformity measures detailed by Romano et al. (2020).

With these ingredients in place, we are in a position to obtain a prediction set $\hat{C}$ for $w_{\text{test}}$, the computation of which is detailed in Algorithm 1. The notation leaves implicit the dependence of the set $\hat{C}$ on the observed data; explicitly, the set $\hat{C}$ is a deterministic function of the dynamic graph $G$, the attributes $(X_w, w \in \nu)$, and the labels $(Y_w, w \in \nu \setminus \nu^{(\text{test})})$.

**Theory.** As mentioned earlier, the transductive regime requires no further assumptions.

**Lemma 1.** *In the transductive regime, the prediction set output by Algorithm 1 is valid, that is,*

$$\mathbb{P}(Y_{w_{\text{test}}} \in \hat{C}) \geq 1 - \alpha.$$

For the semi-inductive regime, we make the following assumptions:

**A1.** *The columns $\mathbf{A}_w, w \in \nu$, attributes $X_w, w \in \nu$, and labels $Y_w, w \in \nu$ are jointly exchangeable, that is, for any permutation $\pi : [m] \to [m]$,*

$$\{\pi G, \pi(X_w, w \in \nu), \pi(Y_w, w \in \nu)\} \triangleq \{G, (X_w, w \in \nu), (Y_w, w \in \nu)\}.$$

---

**Algorithm 1** Split conformal inference

---

**Input:** Dynamic graph $G$, node/time pairs $\nu = \nu^{(\text{training})} \cup \nu^{(\text{validation})} \cup \nu^{(\text{calibration})} \cup \nu^{(\text{test})}$, attributes $(X_w, w \in \nu)$, labels $(Y_w, w \in \nu \setminus \nu^{(\text{test})})$, confidence level $\alpha$, non-conformity function $r$

1: Learn $\hat{\mathbf{V}}^{\text{UNF}}$ based on $\nu^{(\text{training})}$ and $\nu^{(\text{validation})}$
2: Let the initial confidence set be $\hat{C} = \mathcal{Y}$
3: Let $R_w = r(\hat{\mathbf{V}}_w^{\text{UNF}}, Y_w)$ for $w \in \nu^{(\text{calibration})}$
4: Let
$$\hat{q} = \lfloor \alpha(m) \rfloor \text{ largest value in } R_w, w \in \nu^{(\text{calibration})}$$
5: **for** $y \in \mathcal{Y}$ **do**
6:     Let $R_{\text{test}} = r(\hat{\mathbf{V}}_{w_{\text{test}}}^{\text{UNF}}, y)$
7:     Remove $y$ from $\hat{C}$ if $R_{\text{test}} \geq \hat{q}$
8: **end for**

**Output:** Prediction set $\hat{C}$

---

The symbol $\triangleq$ means "equal in distribution". Informally, we can swap columns of $\mathcal{A} = (\mathbf{A}^{(1)}, \ldots, \mathbf{A}^{(T)})$ corresponding to test or calibration pairs, along with the attributes and labels, without changing the likelihood of the data.

Given a permutation $\pi : [m] \to [m]$, we use $\pi G$ to denote a dynamic graph in which the columns of $G$ corresponding to $\nu_1, \ldots, \nu_m$ are permuted according to $\pi$. More precisely, $\pi G$ is the dynamic graph corresponding to the sequence of adjacency matrices $\mathbf{A}^{(1)'}, \ldots, \mathbf{A}^{(T)'}$ obtained as follows. First, copy $\mathbf{A}^{(1)}, \ldots, \mathbf{A}^{(T)}$ to $\mathbf{A}^{(1)'}, \ldots, \mathbf{A}^{(T)'}$. Next, overwrite any column $\nu_\ell \in \nu_1, \ldots, \nu_m$ to $\mathbf{A}'_{\nu_\ell} = \mathbf{A}_{\nu_{\pi^{-1}(\ell)}}$. If $x$ is a sequence of length at least $m$, we write $\pi x$ to denote the same sequence with its first $m$ elements permuted according to $\pi$, that is, $(\pi x)_\ell = x_{\pi^{-1}(\ell)}$ if $\ell \leq m$ and $(\pi x)_\ell = x_\ell$ otherwise.

**A2.** *The GNN $\mathcal{G}$ is permutation equivariant, that is, for any permutation $\pi : [N] \to [N]$ with associated permutation matrix $\Pi$,*

$$\mathcal{G}(\Pi \mathbf{A} \Pi^\top) = \Pi \mathcal{G}(\mathbf{A}).$$

Informally, if we re-order the nodes, attributes and labels, we re-order the resulting embedding.

**Theorem 1.** *Under assumptions 1 and 2, the prediction set output by Algorithm 1 is valid in the semi-inductive regime, that is,*

$$\mathbb{P}(Y_{w_{\text{test}}} \in \hat{C}) \geq 1 - \alpha.$$

**Discussion.** Assumption 2 is relatively mild and roughly satisfied by standard GNN architectures. Assumption 1 is more challenging, and can guide us towards curating a calibration set. For example, if we hypothesise that the $(\mathbf{A}_i^{(t)}, t \in [T])$ are i.i.d. stationary stochastic processes (over time), then we can satisfy Assumption 1 by ensuring $\nu^{(\text{calibration})} \cup \nu^{(\text{test})}$ contains only distinct nodes. The independence hypothesis will typically break in undirected graphs because of the necessary symmetry of each $\mathbf{A}^{(t)}$, and in this case we may also wish to ensure that $\nu^{(\text{calibration})} \cup \nu^{(\text{test})}$ contains only distinct time points. In general, for predicting the label of nodes based on historical information, Assumption 1 should make us wary of global drift, a poor distributional overlap between the test and calibration nodes, and autocorrelation. The semi-inductive regime is where our theory actively requires a dilated unfolding, and where alternative approaches can fail completely; however, even in the transductive regime where our theory covers alternative approaches, the prediction sets from unfolded GNNs (UGNNs) are often smaller.

## 2.1 A Visual Motivation: i.i.d. Draws of a Stochastic Block Model

Consider a simple two-community dynamic stochastic block model (DSBM) (Yang et al., 2011; Xu and Hero, 2014) for undirected graphs. Let

$$\mathbf{B}^{(1)} = \mathbf{B}^{(2)} = \begin{bmatrix} 0.5 & 0.5 \\ 0.5 & 0.9 \end{bmatrix}, \tag{1}$$

be matrices of inter-community edge probabilities, and let $z \in \{1, 2\}^n$ be a community allocation vector. We then draw each symmetric adjacency matrix point as $\mathbf{A}_{ij}^{(1)} \stackrel{\text{ind}}{\sim} \text{Bernoulli}(\mathbf{B}_{z_i, z_j}^{(1)})$ and $\mathbf{A}_{ij}^{(2)} \stackrel{\text{ind}}{\sim} \text{Bernoulli}(\mathbf{B}_{z_i, z_j}^{(2)})$, for $i \leq j$.

A typical way of applying a GNN to a series of graphs is to first stack the adjacency matrices as blocks on the diagonal (Fey and Lenssen, 2019),

$$\mathbf{A}^{\text{BD}} = \begin{bmatrix} \mathbf{A}^{(1)} & & \mathbf{0} \\ & \ddots & \\ \mathbf{0} & & \mathbf{A}^{(T)} \end{bmatrix},$$

from which a dynamic graph embedding is obtained through

$$\hat{\mathbf{V}}^{\text{BD}} = \mathcal{G}\left(\mathbf{A}^{\text{BD}}, X^{\text{BD}}, \{Y_w : w \in \nu^{(\text{training})} \cup \nu^{(\text{validation})}\}\right),$$

where $X^{\text{BD}} \in \mathbb{R}^{nT \times c}$ is the matrix of $c$-dimensional attributes for all node/time pairs. (For networks without attributes, $X^{\text{BD}}$ is set to be the $nT \times nT$ identity matrix). From this representation, it is clear that $\mathbf{A}^{\text{BD}}$ is not exchangeable over time.

Figure 1 plots the representations, $\hat{\mathbf{V}}_{\text{BD}}$ and $\hat{\mathbf{V}}_{\text{UNF}}$ of $\mathbf{A}^{(1)}, \mathbf{A}^{(2)}$ using a GCN (Kipf and Welling, 2016) as $\mathcal{G}$. It is clear that, despite the fact that $\mathbf{A}^{(1)}$ and $\mathbf{A}^{(2)}$ come from the same distribution, the output of the block diagonal GCN appears to encode a change that is not present. We refer to this issue as *temporal shift*. In contrast, the output of UGCN is exchangeable over time and so features no temporal shift.

This example highlights two advantages of unfolded GNN over the standard GNN case. First, as there is no temporal shift, we would expect it to show improved predictive power. Second, exchangeability over time allows for the application of conformal inference across time.

## 3 EXPERIMENTS

We evaluate the performance of UGNNs using four examples, comprising simulated and real data, summarised in Table 1. We will then delve deeper into a particular dataset to show how variation in prediction sets can tell us something about the underlying network dynamics. Figure 2 displays the number of edges in each of the considered datasets over time. The SBM, school and flight data each feature abrupt changes in structure, while the trade data is relatively smooth.

| Dataset | Nodes | Edges | Time points | Classes | Weighted | Directed | Drift |
|---------|-------|-------|-------------|---------|----------|----------|-------|
| SBM | 300 | 38,350 | 8 | 3 | No | No | No |
| School | 232 | 53,172 | 18 | 10 | No | No | No |
| Flight | 2,646 | 1,635,070 | 36 | 46 | Yes | No | No |
| Trade | 255 | 468,245 | 32 | 163 | Yes | Yes | Yes |

Table 1: A summary of considered datasets.

**SBM**. A three-community DSBM with inter-community edge probability matrix

$$\mathbf{B}^{(t)} = \begin{bmatrix} s_1 & 0.02 & 0.02 \\ 0.02 & s_2 & 0.02 \\ 0.02 & 0.02 & s_3 \end{bmatrix},$$

where $s_1$, $s_2$, and $s_3$ represent within-community connection states. Each $s$ can be one of two values: 0.08 or 0.16. We simulate a dynamic network over $T = 8$ time points, corresponding to the $8 = 2^3$ possible combinations, drawing each $\mathbf{A}^{(t)}$ from each unique $\mathbf{B}^{(t)}$ as detailed in Section 2.1. The ordering of these time points is random, leading to dynamic total edge count (Figure 2). The task is to predict the community label of each node.

**School**. A dynamic social network between pupils at a primary school in Lyon, France (Stehlé et al., 2011). Each of the 232 pupils wore a radio identification device such that each interaction, with its

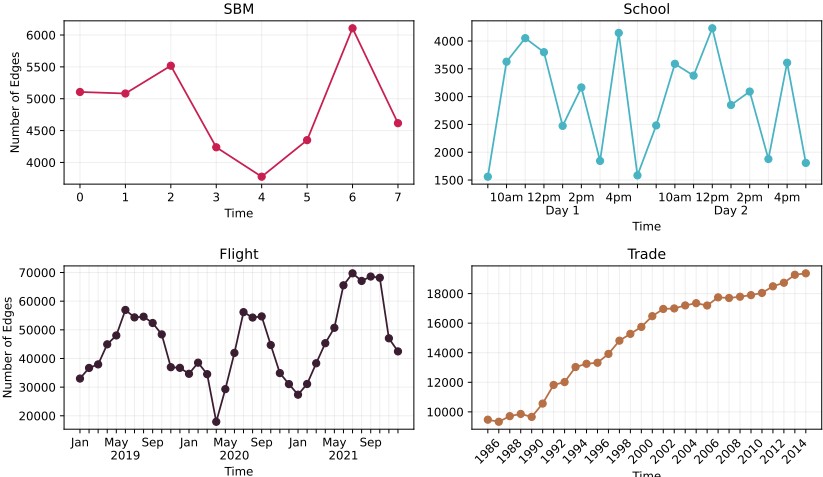

Figure 2: Numbers of edges over time. The School and Flight data show rough periodic/seasonal structure, which the Trade data features drift with the number of edges growing smoothly with time.

timestamp, could be recorded, forming a dynamic network. An interaction was defined by close proximity for 20 seconds. The task here is to predict the classroom allocation of each pupil. This dataset has temporal structure particularly distinguishing class time, where pupils cluster together based on their class (easier), and lunchtime, where the cluster structure breaks down (harder). The data covers two full school days, making it roughly repeating.

**Flight**. The OpenSky dataset tracks the number of flights (edges) between airports (nodes) over each month from the start of 2019 to the end of 2021 (Olive et al., 2022). The task is to predict the country of a given (European only) airport. The network exhibits seasonal and periodic patterns, and features a change in structure when the COVID-19 pandemic hit Europe around March 2020.

**Trade**. An agricultural trade network between members of the United Nations tracked yearly between 1986 and 2016 (MacDonald et al., 2015), which features in the Temporal Graph Benchmark (Huang et al., 2024b). The network is directed and edges are weighted by total trade value. Unlike the other examples, this network exhibits important drift over time. This behaviour is visualised in Figure 2. We consider the goal of predicting the top trade partner for each nation for the next year, a slight deviation from the benchmark where performance is quantified by the Normalized Discounted Cumulative Gain of the top 10 ranked trade partners (less naturally converted into a conformal prediction problem).

## 3.1 EXPERIMENTAL SETUP

On each dataset, we apply GCN (Kipf and Welling, 2016) and GAT (Veličković et al., 2017) to the block diagonal and unfolded matrix structures (referred to as UGCN and UGAT respectively) in both the transductive and the semi-inductive settings. The datasets considered do not have attributes. In the transductive regime, we randomly assign nodes to train, validation, calibration and test sets with ratios 20/10/35/35, regardless of their time point label. Due to the high computational cost of fitting multiple GNNs to quantify randomisation error, we follow (Huang et al., 2024a) by fitting the GNN 10 times, then constructing 100 different permutations of calibration and test sets to allow 1,000 conformal inference instances per model and dataset, and apply Algorithm 2.

For the semi-inductive regime, we apply a similar approach, except that we reserve the *last* 35% of the observation period as the test set, rounded to use an integer number of time points for testing to ensure every node/time pair in the test set is unlabelled. The training, validation and calibration sets are then picked at random, regardless of time point label, from the remaining data with ratios 20/10/35. As the test set is fixed, efficiency saving is not possible and we instead run Algorithm 1 (split conformal) on 50 random data splits.

Prediction sets are computed using the Adaptive Prediction Sets (APS) algorithm (Romano et al., 2020) in both regimes. For each experiment, we return the mean accuracy, coverage and prediction set size across all conformal runs to evaluate the predictive power of each GNN, as well as its conformal performance. To quantify error, we quote the standard deviation of each metric. Code to reproduce all experiments can be found in Appendix A.

## 3.2 RESULTS

**UGNN has higher accuracy in every semi-inductive example and most transductive examples.**
In the transductive regime such advantages are often minor, however in the semi-inductive regime UGNN displays a more significant advantage (Table 2). Block GNN is close to random guessing in the semi-inductive SBM and School examples (having 3 and 10 classes respectively). In contrast, UGNN maintains a strong accuracy in both regimes.

| Methods | SBM | | School | |
|---|---|---|---|---|
| | Trans. | Semi-ind. | Trans. | Semi-ind. |
| Block GCN | $0.964 \pm 0.011$ | $0.334 \pm 0.024$ | $0.856 \pm 0.011$ | $0.116 \pm 0.011$ |
| UGCN | $\mathbf{0.980 \pm 0.004}$ | $\mathbf{0.985 \pm 0.003}$ | $\mathbf{0.924 \pm 0.009}$ | $\mathbf{0.915 \pm 0.013}$ |
| Block GAT | $0.916 \pm 0.028$ | $0.346 \pm 0.024$ | $0.807 \pm 0.016$ | $0.107 \pm 0.024$ |
| UGAT | $\mathbf{0.947 \pm 0.032}$ | $\mathbf{0.969 \pm 0.017}$ | $\mathbf{0.896 \pm 0.016}$ | $\mathbf{0.868 \pm 0.017}$ |
| Methods | Flight | | Trade | |
| | Trans. | Semi-ind. | Trans. | Semi-ind. |
| Block GCN | $0.405 \pm 0.075$ | $0.121 \pm 0.059$ | $\mathbf{0.095 \pm 0.017}$ | $0.049 \pm 0.018$ |
| UGCN | $\mathbf{0.441 \pm 0.069}$ | $\mathbf{0.477 \pm 0.061}$ | $0.082 \pm 0.031$ | $\mathbf{0.050 \pm 0.017}$ |
| Block GAT | $\mathbf{0.417 \pm 0.031}$ | $0.114 \pm 0.061$ | $\mathbf{0.125 \pm 0.017}$ | $0.046 \pm 0.015$ |
| UGAT | $0.408 \pm 0.060$ | $\mathbf{0.427 \pm 0.061}$ | $0.111 \pm 0.010$ | $\mathbf{0.048 \pm 0.015}$ |

Table 2: Accuracy (higher is better) for 2 GNNs (GCN or GAT) under 2 representations (block diagonal adjacency or unfolding) for 4 datasets. Bold values indicate the highest accuracy for a given GNN/representation pair.

**Conformal prediction on UGNN and block GNN is empirically valid for every transductive example.** This confirms (Lemma 1) that exchangeability/label equivariance are not necessary for valid conformal prediction in the transductive regime, as even block GNN (which has no exchangeability properties) is valid here (Table 3). However, UGNN achieves valid coverage with smaller prediction sets in most cases, making it the preferable method (Table 4).

| Methods | SBM | | School | |
|---|---|---|---|---|
| | Trans. | Semi-ind. | Trans. | Semi-ind. |
| Block GCN | $\mathbf{0.901 \pm 0.014}$ | $0.659 \pm 0.045$ | $\mathbf{0.901 \pm 0.012}$ | $0.812 \pm 0.033$ |
| UGCN | $\mathbf{0.901 \pm 0.015}$ | $\mathbf{0.918 \pm 0.025}$ | $\mathbf{0.901 \pm 0.012}$ | $\mathbf{0.924 \pm 0.013}$ |
| Block GAT | $\mathbf{0.901 \pm 0.015}$ | $0.450 \pm 0.154$ | $\mathbf{0.901 \pm 0.012}$ | $0.662 \pm 0.084$ |
| UGAT | $\mathbf{0.901 \pm 0.014}$ | $\mathbf{0.914 \pm 0.022}$ | $\mathbf{0.901 \pm 0.012}$ | $\mathbf{0.909 \pm 0.021}$ |
| Methods | Flight | | Trade | |
| | Trans. | Semi-ind. | Trans. | Semi-ind. |
| Block GCN | $\mathbf{0.900 \pm 0.002}$ | $0.853 \pm 0.013$ | $\mathbf{0.900 \pm 0.009}$ | $0.842 \pm 0.015$ |
| UGCN | $\mathbf{0.900 \pm 0.002}$ | $\mathbf{0.910 \pm 0.003}$ | $\mathbf{0.900 \pm 0.009}$ | $0.847 \pm 0.017$ |
| Block GAT | $\mathbf{0.900 \pm 0.002}$ | $0.862 \pm 0.013$ | $\mathbf{0.900 \pm 0.009}$ | $0.840 \pm 0.021$ |
| UGAT | $\mathbf{0.900 \pm 0.002}$ | $\mathbf{0.906 \pm 0.002}$ | $\mathbf{0.901 \pm 0.009}$ | $0.854 \pm 0.023$ |

Table 3: Coverage (targeted to $\geq 0.9$) for 2 GNNs (GCN or GAT) under 2 representations (block diagonal adjacency or unfolding) for 4 datasets. Bold values indicate valid coverage with target $\geq 0.9$.

**Conformal prediction on UGNN is empirically valid for every semi-inductive example without drift, while block GNN is valid for none.** In all examples, except the trade dataset, UGNN produces valid conformal (Table 3) with similar prediction set sizes to the transductive case (Table 4).

We include the trade data as an example of where UGNN fails to achieve valid coverage in the semi-inductive regime. This failure is due to the drift present in the trade network (Figure 2), which

grows over time (approximately doubling in edges over the whole period). Therefore, the network at the start of this series is not approximately exchangeable with the network at the end of the series. The problem of drift is difficult to handle for UGNN alone, but we hypothesise that downstream methods could be applied to improve coverage in these cases, for example (Barber et al., 2023) and (Clarkson, 2023). We leave this investigation for future work.

Due to the unfolded matrix having double the entries of the block diagonal matrix, the computation times for UGNN were roughly double that of block GNN. The maximum time to train an individual model was around a minute on an AMD Ryzen 5 3600 CPU processor. A full run of experiments on the largest dataset took around 4.5 hours.

In the appendix, we include many further experiments which include a demonstration of validity in the temporal transductive regime (Appendix E), consideration of other score functions for obtaining prediction sets (Appendix G), time-conditional coverage (Appendix D), various other train/valid/calibration/test splits (Appendix F), and use of other GNN architectures (Appendix H). In each case, these further experiments bolster the claims made here; UGNN is superior in both predictive performance and in conformal metrics.

| Methods | SBM | | School | |
|---|---|---|---|---|
| | Trans. | Semi-ind. | Trans. | Semi-ind. |
| Block GCN | **1.258 ± 0.053** | *1.977 ± 0.138* | 4.542 ± 0.167 | *8.079 ± 0.188* |
| UGCN | 1.263 ± 0.206 | **1.097 ± 0.171** | **2.763 ± 0.311** | **3.037 ± 0.251** |
| Block GAT | 1.063 ± 0.180 | *1.320 ± 0.466* | 3.863 ± 0.813 | *6.540 ± 0.830* |
| UGAT | **1.053 ± 0.249** | **1.042 ± 0.201** | **3.552 ± 0.756** | **4.185 ± 1.228** |

| Methods | Flight | | Trade | |
|---|---|---|---|---|
| | Trans. | Semi-ind. | Trans. | Semi-ind. |
| Block GCN | 24.116 ± 2.711 | *23.283 ± 2.285* | **82.556 ± 4.966** | *80.652 ± 6.709* |
| UGCN | **22.369 ± 1.953** | **23.030 ± 2.115** | 86.319 ± 7.520 | *85.006 ± 9.516* |
| Block GAT | 25.173 ± 1.631 | *24.956 ± 1.977* | **87.603 ± 6.945** | *84.708 ± 9.690* |
| UGAT | **24.453 ± 2.368** | **24.451 ± 2.307** | 92.200 ± 7.364 | *90.021 ± 11.592* |

Table 4: Set sizes (lower is better) for 2 GNNs (GCN or GAT) under 2 representations (block adjacency or unfolding) for 4 datasets. Values in bold indicate a smaller set size for a given GNN. Values in italics indicate invalid sets.

## 3.3 TEMPORAL ANALYSIS

The goal of conformal prediction is to provide a notion of uncertainty to an otherwise black-box model. As a problem becomes more difficult, this should be reflected in a larger prediction set size to maintain target coverage. We analyse this behaviour by focusing on the school data example.

Figures 3a and 3d compare the accuracy of both UGAT and block GAT for each time window of the school dataset. During lunchtime, pupils are no longer clustered in their classrooms, making prediction of their class more difficult. Figure 3a confirms a significant drop in accuracy for both models at lunchtime in the transductive case. UGAT also displays this behaviour in the semi-inductive case (Figure 3d), while block GAT's performance is no better than random selection.

At these more difficult lunchtime windows, the prediction set sizes increase for both methods in the transductive case as shown in Figure 3c. UGAT also displays this increase in the semi-inductive regime (Figure 3f), while block GAT does not adapt. This demonstrates that conformal prediction can quantify classification difficulty over time when using a UGNN. Note that while computing accuracy requires the knowledge of test labels to quantify difficulty, conformal prediction of set size does not. Figure 3b confirms that both methods maintain coverage in the transductive case (increasing uncertainty at lunchtimes), and UGAT maintains coverage in the semi-inductive case (Figure 3e). The story is similar for UGCN (Appendix C).

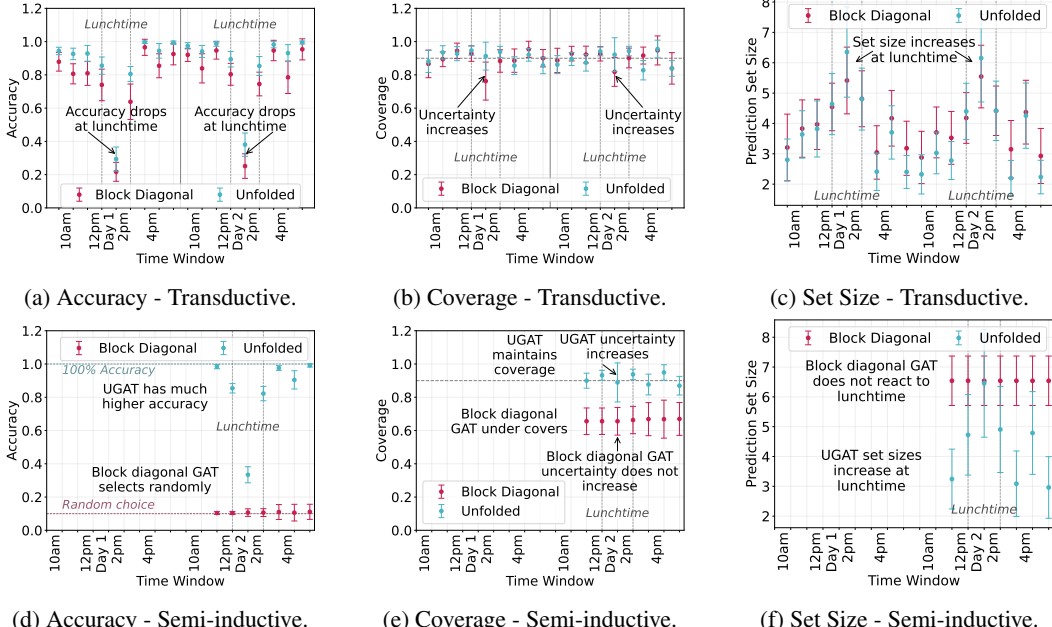

Figure 3: Performance metrics for each time window of the school dataset for unfolded GAT and block diagonal GAT. The prediction task gets more difficult at lunchtime, as shown by the drop in accuracy of both methods in the transductive case. UGAT has marginally better performance in the transductive case and significantly better performance in the semi-inductive case. Prediction set sizes increase at lunchtime, with only UGAT set sizes reacting in the semi-inductive case. Both methods maintain target coverage in the transductive case, with uncertainty increasing at the more difficult lunchtime window. UGAT also maintains target coverage in the semi-inductive case, while block GAT under-covers.

## 4 DISCUSSION

This paper proposes unfolded GNNs, which exhibit exchangeability properties allowing conformal inference in both transductive and semi-inductive regimes. We demonstrate improved predictive performance, coverage, and prediction set size across a range of simulated and real data examples.

As the supplied code will demonstrate, we have intentionally avoided any type of fine-tuning, because the goal here is less prediction accuracy than uncertainty quantification. There are significant opportunities to improve the prediction performance of unfolded GNNs, such as employing architectures better suited to bipartite graphs.

The raw computational challenge of unfolding is modest (a factor of 2) but typical tricks for mega-scale analyses, for example loading time-points into memory independently, cannot be applied. However, the embedding of the behaviour of a node over time ('rows' of $\mathcal{A}$) is in principle amenable to acceleration. Specifically, the behaviour of a node at only a few representative times may be needed to anchor behaviour at a target time. Therefore acceleration of unfolded GNNs to mega-scale is plausible for future work (e.g. Gao et al. (2024a;b)).

There are also opportunities to improve the downstream treatment of embeddings, such as deploying the CF-GNN algorithm (Huang et al., 2024a) to reduce prediction set sizes, or weighting calibration points according to their exchangeability with the test point (Barber et al., 2023; Clarkson, 2023).

Our use of unfolding originates from a body of statistical literature on spectral embedding and tensor decomposition, where it may be useful to observe that inductive embedding is straightforward: the singular vectors provide a projection operator which can be applied to new nodes. Being able to somehow emulate this operation in unfolded GNNs could provide a path towards fully inductive inference.

ACKNOWLEDGEMENTS

Ed Davis gratefully acknowledges support by LV= Insurance (Allianz Group) and the Centre for Doctoral Training in Computational Statistics and Data Science (Compass, EPSRC Grant number EP/S023569/1). Compass is funded by United Kingdom Research and Innovation (UKRI) through the Engineering and Physical Sciences Research Council (EPSRC), `https://www.ukri.org/councils/epsrc`. This work was carried out using the computational facilities of the Advanced Computing Research Centre, University of Bristol - `http://www.bris.ac.uk/acrc/`. PR-D gratefully acknowledges support from the NeST programme grant, EPSRC grant number EP/X002195/1.

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

## A  CODE AVAILABILITY

Python code for reproducing experiments is available at `https://github.com/edwarddavis1/valid_conformal_for_dynamic_gnn`.

## B  Supporting theory for Section 2

### B.1  Transductive theory

Our arguments are similar in style to (Barber et al., 2023), showing validity of full conformal inference, and then split conformal as a special case of full conformal.

We observe $Y_w$ for all $w \in \nu$ apart from $Y_{\nu_K}$, where the query point $K$ is chosen independently and uniformly at random among $\{1, \ldots, m\}$.

#### B.1.1  Full conformal prediction

Suppose we have access to an algorithm $\mathcal{A}(g, x, y)$ which takes as input:

1. $g$, a dynamic graph on $n$ nodes and $T$ time points;
2. $x$, a $|\nu|$-long sequence of feature vectors;
3. $y$, a $|\nu|$-long sequence of labels;

and returns a sequence $r = (r_1, \ldots, r_m)$ of real values, which will act as non-conformity scores.

---

**Algorithm 2** Full conformal inference

**Input:** Dynamic graph $G$, index set $\nu$, attributes $(X_w, w \in \nu)$, labels $(Y_w, w \in \nu \setminus \nu_K)$, confidence level $\alpha$, algorithm $\mathcal{A}$, query point $K$

1: Let $\hat{C} = \mathcal{Y}$
2: **for** $y \in \mathcal{Y}$ **do**
3:    Let
$$X = (X_{\nu_1}, \ldots, X_{\nu_{|\nu|}}); \quad Y^+ = (Y_{\nu_1}, \ldots, Y_{\nu_{K-1}}, y, Y_{\nu_{K+1}}, \ldots, Y_{\nu_{|\nu|}}).$$
4:    Compute
$$(R_1, \ldots, R_m) = \mathcal{A}(G, X, Y^+).$$
5:    Remove $y$ from $\hat{C}$ if
$$R_K \text{ among } \lfloor \alpha(m) \rfloor \text{ largest of } R_1, \ldots, R_m.$$
6: **end for**

**Output:** Prediction set $\hat{C}$

---

**Lemma 2.** *The prediction set output by Algorithm 2 is valid.*

*Proof.* Because $K \overset{ind}{\sim} \text{uniform}([m])$, the event
$$E = \text{``}R_K \text{ among } \lfloor \alpha(m) \rfloor \text{ largest of } R_1, \ldots, R_m \text{''},$$
occurs with probability
$$\mathbb{P}(E) \le \alpha.$$
But $E$ occurs if and only if $Y_{\nu_K} \notin \hat{C}$. Therefore,
$$\mathbb{P}(Y_{\nu_K} \in \hat{C}) = 1 - \mathbb{P}(Y_{\nu_K} \notin \hat{C}) = 1 - \mathbb{P}(E) \ge 1 - \alpha.$$
$\square$

What this formalism makes clear is that the validity of full conformal inference in a transductive setting has nothing to do with exchangeability or any form of symmetry in $\mathcal{A}$ (neither of which were assumed).

---

**Algorithm 3** Split conformal inference (abstract version)

---

**Input:** Dynamic graph $G$, index set $\nu$, attributes $(X_w, w \in \nu)$, labels $(Y_w, w \in \nu \setminus \nu_K)$, confidence level $\alpha$, prediction function $\hat{f}$, non-conformity function $r$, query point $K$

1: Let $\hat{C} = \mathcal{Y}$
2: Let $R_i = r(\hat{f}(\nu_i), Y_{\nu_i})$ for $i \in [m] \setminus K$
3: Let
$$\hat{q} = \lfloor \alpha(m) \rfloor \text{ largest value in } R_i, i \in [m] \setminus K$$
4: **for** $y \in \mathcal{Y}$ **do**
5:     Let $R_K = r(\hat{f}(\nu_K), y)$
6:     Remove $y$ from $\hat{C}$ if $R_K \geq \hat{q}$
7: **end for**

**Output:** Prediction set $\hat{C}$

---

### B.1.2 Split conformal prediction

Suppose we construct $\mathcal{A}$ according to a split training/calibration routine, as follows. First, using $g, \nu$, the feature vectors $(X_w, w \in \nu)$, and the labels $Y_{\nu_\ell}, \ell > m$, learn a prediction function $\hat{f} : \nu \to \mathbb{R}^d$.

The vector $\hat{f}(\nu_\ell)$ might, for example, relate to predicted class probabilities via

$$\hat{\mathbb{P}}(Y_{\nu_\ell} = k) = \left[ \text{softmax}(\hat{f}(\nu_\ell)_1, \ldots, \hat{f}(\nu_\ell)_d) \right]_k,$$

in which case $\hat{f}(\nu_\ell)$ is often known as an embedding.

Second, for a non-conformity function $r : \mathbb{R}^d \times \mathcal{Y} \to \mathbb{R}$, a dynamic graph $g$, and $x, y$ of length $|\nu|$, define
$$[\mathcal{A}(g, x, y)]_\ell = r(\hat{f}(\nu_\ell), y_\ell), \quad \ell \in [m].$$

Then Algorithm 2 can potentially be made much more efficient, as shown in Algorithm 1. This is nothing but the usual split conformal inference algorithm, where the labels $Y_{\nu_\ell}, \ell > m$ are being used for training, and $Y_{\nu_\ell}, \ell \in [m] \setminus K$ for calibration.

Thus split conformal inference is a special case of full conformal inference, in which special structure is imposed on $\mathcal{A}$, and must therefore inherit its general validity (Lemma 1).

### B.2 Semi-inductive theory

Otherwise keeping the same setup, suppose $K$ is not uniformly chosen but is instead fixed, to (say) $K = 1$.

**A3.** *The algorithm $\mathcal{A}(g, x, y)$ is symmetric in its input. For any permutation $\pi : \nu \to \nu$,*

$$\mathcal{A}(\pi g, \pi x, \pi y) = \pi \mathcal{A}(g, x, y)$$

**Theorem 2.** *Under assumptions 1 and 3, the prediction set output by Algorithm 2 (and so Algorithm 3) is valid even if $K = 1$ deterministically.*

*Proof.* The combination of assumptions 1 and 3 ensures that $R_1, \ldots, R_m$ are exchangeable, and therefore the event
$$E = \text{``}R_1 \text{ among } \lfloor \alpha(m) \rfloor \text{ largest of } R_1, \ldots, R_m\text{''},$$
occurs with probability
$$\mathbb{P}(E) \leq \alpha.$$
But $E$ occurs if and only if $Y_{\nu_1} \notin \hat{C}$. Therefore,
$$\mathbb{P}(Y_{\nu_1} \in \hat{C}) = 1 - \mathbb{P}(Y_{\nu_K} \notin \hat{C}) = 1 - \mathbb{P}(E) \geq 1 - \alpha.$$

$\square$

To map Theorem 2 to Theorem 1, we observe that Assumptions 1 and 1 are the same (the latter a detailed version of the former); and that inputting $\mathbf{A}^{\text{UNF}}$ to a GNN satisfying Assumption 2, along with the attributes $X_w, w \in \nu$ and training labels $Y_w, w \in \nu^{(\text{training})}$, results in an $\mathcal{A}$ satisfying Assumption 3.

## C  TEMPORAL ANALYSIS USING GCN

In this section, we present a temporal analysis of the school data example using GCN instead of GAT as the GNN model. We see similar conclusions to those stated in Section 3. The accuracy of UGCN is slightly higher at every time point in the transductive case and much higher in the semi-inductive case. Block GCN is essentially guessing randomly in the semi-inductive case. The prediction sets for UGCN are always smaller than those of block GCN, with UGCN's set sizes also reacting to the difficulty problem in the semi-inductive regime. However, neither method achieves valid coverage of every time point in the transductive regime, in contrast to GAT. UGCN also under-covers the first test point in the semi-inductive case (unlike UGAT), while block GCN continually under-covers here.

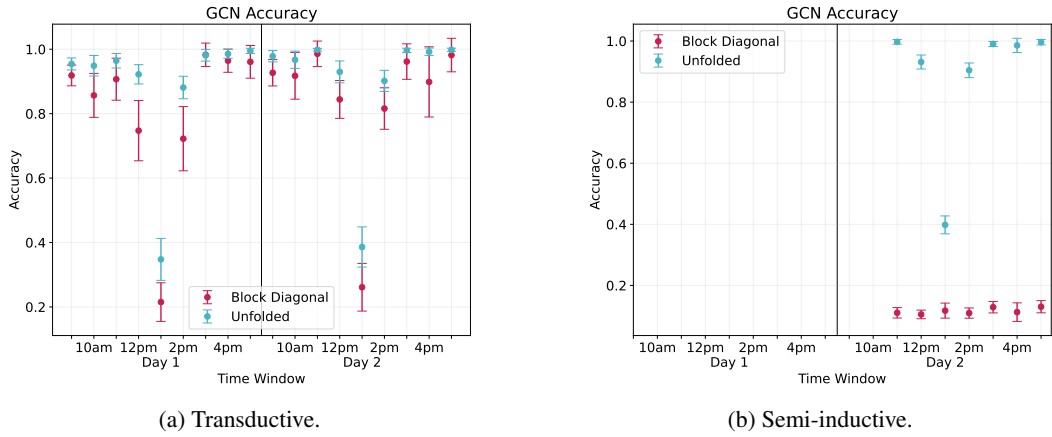

(a) Transductive.                                    (b) Semi-inductive.

Figure 4: Prediction accuracy for each time window of the school dataset for unfolded GCN and block diagonal GCN. The prediction task gets more difficult at lunchtime, as shown by the drop in accuracy of both methods in the transductive case. UGCN has marginally better performance in the transductive case and significantly better performance in the semi-inductive case.

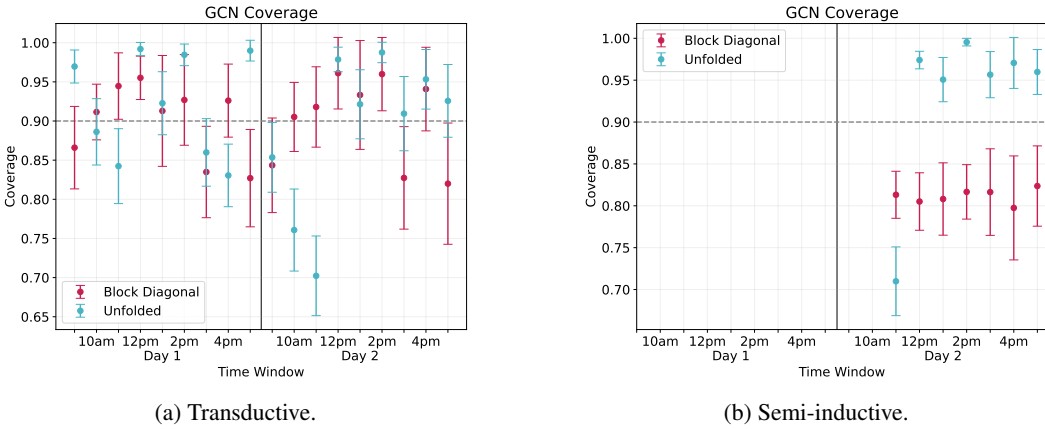

(a) Transductive.                                    (b) Semi-inductive.

Figure 5: Coverage for each time window of the school dataset for unfolded GCN and block diagonal GCN. Both methods maintain target coverage on average in the transductive case, but not at every point in time. In the semi-inductive case, block GCN under-covers continuously and UGCN under-covers for the first test point.

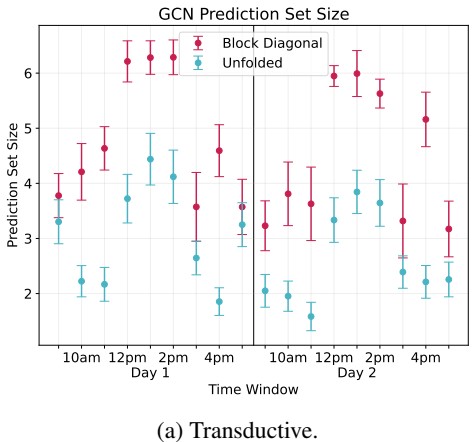
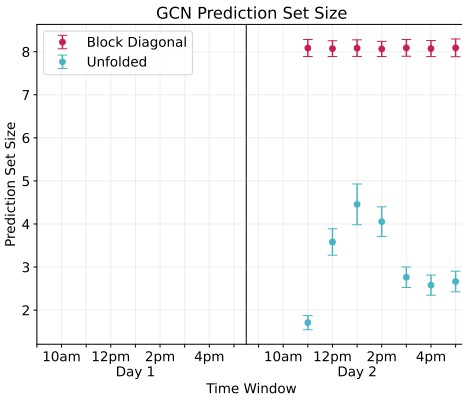

(a) Transductive.              (b) Semi-inductive.

Figure 6: Prediction set sizes for each time window of the school dataset for unfolded GCN and block diagonal GCN. As the prediction task gets more difficult at lunchtime the prediction set sizes increase. Only the UGCN set sizes react to lunchtime in the semi-inductive case.

## D  TIME-CONDITIONAL COVERAGE

When evaluating a conformal procedure, it is common to consider conditional coverage (Angelopoulos and Bates, 2021; Huang et al., 2024a; Lei and Wasserman, 2014). A procedure with conditional coverage would be very adaptable, as it would return prediction sets with coverage $1 - \alpha$ conditional on the test point. While this property is desirable, it is known to be impossible to achieve exact distribution-free conditional coverage (Lei and Wasserman, 2014). Despite this, it is common to evaluate approximate conditional coverage by looking at the worst-case coverage across certain features or groups (Angelopoulos and Bates, 2021). For the purposes of this paper, we evaluate the *time-conditional* coverage, in other words, the worst-case coverage for a single time point out of all time points in the series.

Table 5 displays the time-conditional coverage for each method. While UGAT is often valid, it is not generally true that either block GNN or UGNN achieves validity here. The reason for this is again rooted in data drift. As no two time points are exactly exchangeable, no two *embedded* time points will be exactly exchangeable. This then translates to variation in coverage between time points. In the marginal coverage case, variation is fine as long as the average coverage is on target. However, under this metric, any variation causes a decrease.

Despite this, let us consider an example with no drift between time points. Consider another three-community DSBM with inter-community edge probability matrix

$$\mathbf{B} = \begin{bmatrix} 0.16 & 0.02 & 0.02 \\ 0.02 & 0.08 & 0.02 \\ 0.02 & 0.02 & 0.16 \end{bmatrix},$$

and draw each $\mathbf{A}^{(t)}$ from $\mathbf{B}$. This means that the series is simply a list of i.i.d. draws and so features no data drift. Table 6 displays the result of running our experiments on this data. As expected, each method achieved value time-conditional coverage in the transductive and temporal transductive regimes as drift between time points is no longer a problem. However, even with such a basic data example, it is only the UGNNs that are able to achieve valid temporal-conditional coverage here.

| Methods | SBM | | School | |
|---|---|---|---|---|
| | Trans. | Semi-ind. | Trans. | Semi-ind. |
| Block GCN | $0.783 \pm 0.091$ | $0.645 \pm 0.055$ | $0.820 \pm 0.077$ | $0.797 \pm 0.062$ |
| UGCN | $0.806 \pm 0.044$ | $0.869 \pm 0.030$ | $0.702 \pm 0.051$ | $0.710 \pm 0.041$ |
| Block GAT | $\mathbf{0.856 \pm 0.068}$ | $0.450 \pm 0.154$ | $0.763 \pm 0.114$ | $0.656 \pm 0.083$ |
| UGAT | $\mathbf{0.877 \pm 0.040}$ | $\mathbf{0.907 \pm 0.031}$ | $0.828 \pm 0.058$ | $\mathbf{0.870 \pm 0.056}$ |

| Methods | Flight | | Trade | |
|---|---|---|---|---|
| | Trans. | Semi-ind. | Trans. | Semi-ind. |
| Block GCN | $0.881 \pm 0.011$ | $0.852 \pm 0.013$ | $0.801 \pm 0.064$ | $0.767 \pm 0.044$ |
| UGCN | $0.877 \pm 0.011$ | $0.894 \pm 0.004$ | $0.800 \pm 0.094$ | $0.774 \pm 0.056$ |
| Block GAT | $0.885 \pm 0.011$ | $0.861 \pm 0.014$ | $0.822 \pm 0.061$ | $0.776 \pm 0.046$ |
| UGAT | $\mathbf{0.889 \pm 0.009}$ | $\mathbf{0.896 \pm 0.004}$ | $0.784 \pm 0.076$ | $0.792 \pm 0.065$ |

Table 5: Time-conditional coverage over time with target $\geq 0.90$. Values in bold indicate valid coverage.

| Methods | SBM (i.i.d.) | | |
|---|---|---|---|
| | Trans. | Temp. Trans. | Semi-Ind. |
| Block GCN | $\mathbf{0.875 \pm 0.063}$ | $\mathbf{0.982 \pm 0.018}$ | $0.653 \pm 0.072$ |
| UGCN | $\mathbf{0.880 \pm 0.031}$ | $\mathbf{0.881 \pm 0.048}$ | $\mathbf{0.891 \pm 0.010}$ |
| Block GAT | $\mathbf{0.862 \pm 0.079}$ | $\mathbf{0.976 \pm 0.022}$ | $0.449 \pm 0.158$ |
| UGAT | $\mathbf{0.883 \pm 0.031}$ | $\mathbf{0.887 \pm 0.045}$ | $\mathbf{0.885 \pm 0.016}$ |

Table 6: Time-conditional coverage on an i.i.d. version of the SBM experiment. Bold values indicate validity with target $\geq 0.90$.

## E  TEMPORAL TRANSDUCTIVE

As mentioned in Section 2, and visualised in Figure 1b (bottom), the temporal transductive regime represents the case where the train and validation sets are selected randomly from data at times $t < T$, and calibration and test sets are selected randomly from $t \geq T$. This regime exists as a more practical version of the standard transductive regime, despite, theoretically, requiring no extra assumptions for valid conformal sets. However, this regime presents more of a challenge in comparison to the standard transductive case as we require a model that is trained on $t < T$ to generalise to $t \geq T$.

Table 7 displays the results of running our experiments in this regime. As predicted by our theory, and similarly to the transductive experiments, every method (including the block diagonal methods) achieves valid coverage here. However, as only the unfolded methods are exchangeable over time, the UGNN methods have a significantly higher accuracy on the datasets with out drift. Due to this, the average prediction set size is also reduced for the UGNNs.

| Methods | SBM | School | Flight | Trade |
|---|---|---|---|---|
| Block GCN | $0.318 \pm 0.036$ | $0.118 \pm 0.025$ | $0.103 \pm 0.063$ | $\mathbf{0.052 \pm 0.014}$ |
| UGCN | $\mathbf{0.984 \pm 0.010}$ | $\mathbf{0.925 \pm 0.020}$ | $\mathbf{0.464 \pm 0.075}$ | $0.043 \pm 0.021$ |
| Block GAT | $0.355 \pm 0.046$ | $0.104 \pm 0.032$ | $0.106 \pm 0.064$ | $0.049 \pm 0.015$ |
| UGAT | $\mathbf{0.980 \pm 0.012}$ | $\mathbf{0.903 \pm 0.028}$ | $\mathbf{0.420 \pm 0.064}$ | $\mathbf{0.051 \pm 0.016}$ |

(a) Accuracy. Bold values indicate higher accuracy.

| Methods | SBM | School | Flight | Trade |
|---|---|---|---|---|
| Block GCN | $\mathbf{0.985 \pm 0.011}$ | $\mathbf{0.911 \pm 0.026}$ | $\mathbf{0.900 \pm 0.005}$ | $\mathbf{0.901 \pm 0.014}$ |
| UGCN | $\mathbf{0.904 \pm 0.028}$ | $\mathbf{0.901 \pm 0.027}$ | $\mathbf{0.900 \pm 0.005}$ | $\mathbf{0.901 \pm 0.015}$ |
| Block GAT | $\mathbf{0.977 \pm 0.014}$ | $\mathbf{0.931 \pm 0.033}$ | $\mathbf{0.900 \pm 0.005}$ | $\mathbf{0.901 \pm 0.015}$ |
| UGAT | $\mathbf{0.905 \pm 0.027}$ | $\mathbf{0.902 \pm 0.026}$ | $\mathbf{0.900 \pm 0.005}$ | $\mathbf{0.901 \pm 0.015}$ |

(b) Coverage. Bold values indicate validity with target $\geq 0.90$.

| Methods | SBM | School | Flight | Trade |
|---|---|---|---|---|
| Block GCN | $2.952 \pm 0.018$ | $9.069 \pm 0.186$ | $30.075 \pm 2.783$ | $\mathbf{111.315 \pm 10.401}$ |
| UGCN | $\mathbf{1.170 \pm 0.199}$ | $\mathbf{3.718 \pm 0.259}$ | $\mathbf{21.466 \pm 2.226}$ | $112.427 \pm 8.300$ |
| Block GAT | $2.933 \pm 0.031$ | $9.153 \pm 0.322$ | $30.842 \pm 2.519$ | $115.805 \pm 8.289$ |
| UGAT | $\mathbf{0.933 \pm 0.036}$ | $\mathbf{4.488 \pm 0.698}$ | $\mathbf{23.626 \pm 2.399}$ | $\mathbf{111.892 \pm 7.711}$ |

(c) Average set size. Bold values indicate smaller set sizes.

| Methods | SBM | School | Flight | Trade |
|---|---|---|---|---|
| Block GCN | $\mathbf{0.985 \pm 0.017}$ | $\mathbf{0.898 \pm 0.072}$ | $\mathbf{0.899 \pm 0.017}$ | $\mathbf{0.858 \pm 0.055}$ |
| UGCN | $\mathbf{0.858 \pm 0.080}$ | $0.654 \pm 0.104$ | $\mathbf{0.884 \pm 0.017}$ | $0.842 \pm 0.049$ |
| Block GAT | $\mathbf{0.975 \pm 0.024}$ | $\mathbf{0.914 \pm 0.083}$ | $\mathbf{0.898 \pm 0.017}$ | $\mathbf{0.850 \pm 0.055}$ |
| UGAT | $\mathbf{0.870 \pm 0.058}$ | $\mathbf{0.832 \pm 0.084}$ | $\mathbf{0.891 \pm 0.017}$ | $\mathbf{0.844 \pm 0.068}$ |

(d) Time-conditional coverage. Bold values indicate validity with target $\geq 0.90$.

Table 7: Experiments run in the temporal transductive regime.

## F    REPEAT EXPERIMENTS WITH DIFFERENT DATA SPLITS

Throughout this work, we have used a standard 20/10/35/35 train/validation/calibration/testing data split. To explore the sensitivity of our results to data split selection, we carry out a repeat of our experiments using six different data split ratios. For the sake of computational efficiency, we repeat these experiments using only the school dataset.

Across the different data splits, we almost always achieve the same results as those in the main text. While there is one case of UGCN just missing out on valid coverage, and a few cases of block GCN achieving valid coverage in the semi-inductive regime, we never see block GNN achieving valid coverage when UGNN does not. Therefore, UGNN appears to be the favoured method, regardless of data split choice.

| Embedding | School | | |
|---|---|---|---|
| | Trans. | Semi-ind. | Temp. Trans. |
| Block GCN | **0.901 ± 0.014** | 0.833 ± 0.046 | **0.909 ± 0.019** |
| UGCN | **0.901 ± 0.014** | **0.943 ± 0.018** | **0.903 ± 0.014** |
| Block GAT | **0.902 ± 0.014** | 0.681 ± 0.102 | **0.916 ± 0.026** |
| UGAT | **0.901 ± 0.014** | **0.912 ± 0.024** | **0.901 ± 0.014** |

(a) Coverage for the School experiment with data split 25/25/25/25.

| Embedding | School | | |
|---|---|---|---|
| | Trans. | Semi-ind. | Temp. Trans. |
| Block GCN | **0.903 ± 0.015** | 0.826 ± 0.075 | **0.913 ± 0.020** |
| UGCN | **0.902 ± 0.016** | **0.939 ± 0.020** | **0.902 ± 0.016** |
| Block GAT | **0.904 ± 0.015** | 0.692 ± 0.109 | **0.926 ± 0.029** |
| UGAT | **0.903 ± 0.016** | **0.901 ± 0.035** | **0.902 ± 0.015** |

(b) Coverage for the School experiment with data split 50/10/20/20.

| Embedding | School | | |
|---|---|---|---|
| | Trans. | Semi-ind. | Temp. Trans. |
| Block GCN | **0.901 ± 0.011** | 0.822 ± 0.055 | **0.905 ± 0.016** |
| UGCN | **0.900 ± 0.011** | 0.867 ± 0.022 | **0.899 ± 0.011** |
| Block GAT | **0.901 ± 0.011** | 0.628 ± 0.099 | **0.909 ± 0.017** |
| UGAT | **0.901 ± 0.011** | **0.909 ± 0.021** | **0.901 ± 0.011** |

(c) Coverage for the School experiment with data split 10/10/40/40.

| Embedding | School | | |
|---|---|---|---|
| | Trans. | Semi-ind. | Temp. Trans. |
| Block GCN | **0.901 ± 0.012** | 0.823 ± 0.049 | **0.906 ± 0.017** |
| UGCN | **0.901 ± 0.012** | **0.889 ± 0.020** | **0.903 ± 0.012** |
| Block GAT | **0.901 ± 0.012** | 0.654 ± 0.093 | **0.914 ± 0.022** |
| UGAT | **0.901 ± 0.012** | **0.907 ± 0.024** | **0.901 ± 0.012** |

(d) Coverage for the School experiment with data split 20/10/35/35.

| Embedding | School | | |
|---|---|---|---|
| | Trans. | Semi-ind. | Temp. Trans. |
| Block GCN | **0.906 ± 0.022** | 0.845 ± 0.070 | **0.918 ± 0.031** |
| UGCN | **0.905 ± 0.022** | **0.903 ± 0.030** | **0.906 ± 0.026** |
| Block GAT | **0.906 ± 0.022** | 0.705 ± 0.115 | **0.946 ± 0.046** |
| UGAT | **0.906 ± 0.022** | 0.876 ± 0.042 | **0.907 ± 0.026** |

(e) Coverage for the School experiment with data split 50/30/10/10.

| Embedding | School | | |
|---|---|---|---|
| | Trans. | Semi-ind. | Temp. Trans. |
| Block GCN | **0.901 ± 0.013** | 0.870 ± 0.039 | **0.910 ± 0.017** |
| UGCN | **0.901 ± 0.013** | 0.897 ± 0.020 | **0.902 ± 0.014** |
| Block GAT | **0.901 ± 0.013** | 0.786 ± 0.069 | **0.908 ± 0.019** |
| UGAT | **0.902 ± 0.013** | **0.908 ± 0.023** | **0.901 ± 0.013** |

(f) Coverage for the School experiment with data split 5/35/30/30.

Table 8: Coverage for repeated runs of the School experiment with various data splits. Bold values indicate valid coverage with target $\geq 0.9$.

| Embedding | School | | |
|---|---|---|---|
| | Trans. | Semi-ind. | Temp. Trans. |
| Block GCN | $0.888 \pm 0.013$ | $0.109 \pm 0.025$ | $0.112 \pm 0.010$ |
| UGCN | $\mathbf{0.935 \pm 0.007}$ | $\mathbf{0.896 \pm 0.010}$ | $\mathbf{0.938 \pm 0.006}$ |
| Block GAT | $0.834 \pm 0.021$ | $0.108 \pm 0.028$ | $0.106 \pm 0.018$ |
| UGAT | $\mathbf{0.908 \pm 0.019}$ | $\mathbf{0.883 \pm 0.014}$ | $\mathbf{0.904 \pm 0.024}$ |

(a) Accuracy for the School experiment with data split 25/25/25/25.

| Embedding | School | | |
|---|---|---|---|
| | Trans. | Semi-ind. | Temp. Trans. |
| Block GCN | $0.911 \pm 0.010$ | $0.108 \pm 0.035$ | $0.118 \pm 0.015$ |
| UGCN | $\mathbf{0.931 \pm 0.010}$ | $\mathbf{0.971 \pm 0.008}$ | $\mathbf{0.923 \pm 0.009}$ |
| Block GAT | $0.880 \pm 0.016$ | $0.111 \pm 0.034$ | $0.095 \pm 0.032$ |
| UGAT | $\mathbf{0.912 \pm 0.017}$ | $\mathbf{0.962 \pm 0.010}$ | $\mathbf{0.908 \pm 0.014}$ |

(b) Accuracy for the School experiment with data split 50/10/20/20.

| Embedding | School | | |
|---|---|---|---|
| | Trans. | Semi-ind. | Temp. Trans. |
| Block GCN | $0.787 \pm 0.020$ | $0.105 \pm 0.019$ | $0.110 \pm 0.010$ |
| UGCN | $\mathbf{0.928 \pm 0.009}$ | $\mathbf{0.913 \pm 0.014}$ | $\mathbf{0.924 \pm 0.006}$ |
| Block GAT | $0.697 \pm 0.031$ | $0.102 \pm 0.023$ | $0.108 \pm 0.015$ |
| UGAT | $\mathbf{0.880 \pm 0.019}$ | $\mathbf{0.903 \pm 0.016}$ | $\mathbf{0.876 \pm 0.014}$ |

(c) Accuracy for the School experiment with data split 10/10/40/40.

| Embedding | School | | |
|---|---|---|---|
| | Trans. | Semi-ind. | Temp. Trans. |
| Block GCN | $0.871 \pm 0.010$ | $0.105 \pm 0.024$ | $0.113 \pm 0.009$ |
| UGCN | $\mathbf{0.933 \pm 0.006}$ | $\mathbf{0.913 \pm 0.009}$ | $\mathbf{0.953 \pm 0.006}$ |
| Block GAT | $0.808 \pm 0.018$ | $0.107 \pm 0.025$ | $0.109 \pm 0.010$ |
| UGAT | $\mathbf{0.901 \pm 0.021}$ | $\mathbf{0.899 \pm 0.012}$ | $\mathbf{0.910 \pm 0.014}$ |

(d) Accuracy for the School experiment with data split 20/10/35/35.

| Embedding | School | | |
|---|---|---|---|
| | Trans. | Semi-ind. | Temp. Trans. |
| Block GCN | $0.910 \pm 0.016$ | $0.113 \pm 0.038$ | $0.135 \pm 0.026$ |
| UGCN | $\mathbf{0.934 \pm 0.014}$ | $\mathbf{0.996 \pm 0.006}$ | $\mathbf{0.998 \pm 0.002}$ |
| Block GAT | $0.881 \pm 0.017$ | $0.116 \pm 0.035$ | $0.114 \pm 0.041$ |
| UGAT | $\mathbf{0.913 \pm 0.018}$ | $\mathbf{0.991 \pm 0.007}$ | $\mathbf{0.979 \pm 0.013}$ |

(e) Accuracy for the School experiment with data split 50/30/10/10.

| Embedding | School | | |
|---|---|---|---|
| | Trans. | Semi-ind. | Temp. Trans. |
| Block GCN | $0.583 \pm 0.031$ | $0.105 \pm 0.022$ | $0.105 \pm 0.011$ |
| UGCN | $\mathbf{0.918 \pm 0.026}$ | $\mathbf{0.897 \pm 0.022}$ | $\mathbf{0.939 \pm 0.009}$ |
| Block GAT | $0.543 \pm 0.030$ | $0.106 \pm 0.025$ | $0.111 \pm 0.010$ |
| UGAT | $\mathbf{0.838 \pm 0.031}$ | $\mathbf{0.874 \pm 0.022}$ | $\mathbf{0.877 \pm 0.032}$ |

(f) Accuracy for the School experiment with data split 5/35/30/30.

Table 9: Accuracy values (higher is better) for repeated runs of the School experiment with various data splits. Bold values indicate the highest accuracy for a given GNN/representation pair.

## G   REPEAT EXPERIMENTS USING DIFFERENT SCORE FUNCTIONS

Up to now, we have used adaptive prediction sets (APS) (Romano et al., 2020) to get our prediction sets. However, our theoretical results are not limited to a single choice of conformal procedure. To highlight this, we include experiments where we use the APS, RAPS (Angelopoulos et al., 2020) and SAPS (Huang et al., 2023) methods to compute prediction sets.

As both RAPS and SAPS require the choice of a hyperparameter, we follow Huang et al. (2023) in holding out 20% of the calibration set to select the best hyperparameter for these methods. As the school dataset is the smallest real-world dataset considered in this work, we run these extra experiments on this dataset alone.

In almost all cases, both block GNN and UGNN achieve valid coverage in the two transductive regimes, with UGNN also mostly being valid in the semi-inductive regime.

| Embedding | School | | |
|---|---|---|---|
| | Trans. | Semi-ind. | Temp. Trans. |
| Block GCN | **0.901 ± 0.012** | 0.821 ± 0.049 | **0.913 ± 0.024** |
| UGCN | **0.901 ± 0.012** | **0.889 ± 0.020** | **0.902 ± 0.016** |
| Block GAT | **0.901 ± 0.012** | 0.665 ± 0.097 | **0.922 ± 0.027** |
| UGAT | **0.901 ± 0.012** | **0.907 ± 0.023** | **0.903 ± 0.016** |

(a) Coverage for the School experiment using APS.

| Embedding | School | | |
|---|---|---|---|
| | Trans. | Semi-ind. | Temp. Trans. |
| Block GCN | **0.902 ± 0.013** | 0.821 ± 0.049 | **0.915 ± 0.027** |
| UGCN | **0.901 ± 0.012** | **0.890 ± 0.020** | **0.903 ± 0.017** |
| Block GAT | **0.902 ± 0.012** | 0.665 ± 0.096 | **0.921 ± 0.029** |
| UGAT | **0.901 ± 0.013** | **0.907 ± 0.024** | **0.904 ± 0.017** |

(b) Coverage for the School experiment using RAPS.

| Embedding | School | | |
|---|---|---|---|
| | Trans. | Semi-ind. | Temp. Trans. |
| Block GCN | **0.891 ± 0.014** | 0.817 ± 0.050 | 0.226 ± 0.018 |
| UGCN | **0.897 ± 0.013** | 0.875 ± 0.023 | **0.901 ± 0.017** |
| Block GAT | 0.855 ± 0.022 | 0.651 ± 0.098 | 0.169 ± 0.097 |
| UGAT | **0.895 ± 0.017** | **0.900 ± 0.025** | **0.898 ± 0.020** |

(c) Coverage for the School experiment using SAPS.

Table 10: Coverage values for of the School experiment using different conformal score functions. Values in bold indicate valid coverage with target $\geq 0.90$.

| Embedding | School | | |
|---|---|---|---|
| | Trans. | Semi-ind. | Temp. Trans. |
| Block GCN | $5.195 \pm 0.156$ | *$8.174 \pm 0.386$* | $9.121 \pm 0.254$ |
| UGCN | **$3.541 \pm 0.314$** | **$2.714 \pm 0.326$** | **$3.719 \pm 0.239$** |
| Block GAT | **$4.367 \pm 0.798$** | *$6.567 \pm 0.891$* | $9.124 \pm 0.310$ |
| UGAT | $4.480 \pm 0.882$ | **$3.024 \pm 0.508$** | **$4.503 \pm 0.683$** |

(a) Average set sizes for the School experiment using APS.

| Embedding | School | | |
|---|---|---|---|
| | Trans. | Semi-ind. | Temp. Trans. |
| Block GCN | $5.205 \pm 0.169$ | *$8.166 \pm 0.393$* | $9.135 \pm 0.273$ |
| UGCN | **$3.549 \pm 0.318$** | **$2.728 \pm 0.336$** | **$3.735 \pm 0.250$** |
| Block GAT | **$4.377 \pm 0.802$** | *$6.565 \pm 0.887$* | $9.117 \pm 0.322$ |
| UGAT | $4.486 \pm 0.891$ | **$3.029 \pm 0.501$** | **$4.516 \pm 0.696$** |

(b) Average set sizes for the School experiment using RAPS.

| Embedding | School | | |
|---|---|---|---|
| | Trans. | Semi-ind. | Temp. Trans. |
| Block GCN | $5.044 \pm 0.162$ | *$8.130 \pm 0.400$* | *$1.972 \pm 0.030$* |
| UGCN | **$3.489 \pm 0.329$** | *$2.593 \pm 0.355$* | **$3.719 \pm 0.254$** |
| Block GAT | *$3.755 \pm 0.770$* | *$6.419 \pm 0.936$* | *$1.570 \pm 0.886$* |
| UGAT | **$4.451 \pm 0.954$** | **$2.982 \pm 0.519$** | **$4.483 \pm 0.740$** |

(c) Average set sizes for the School experiment using SAPS.

Table 11: The average prediction set size for each method using APS, RAPS and SAPS on the school dataset. Values in bold indicate the smallest valid prediction set size for a given GNN architecture. Values in italics are the set sizes for invalid prediction sets.

## H    REPEAT EXPERIMENTS USING DIFFERENT GNNS

In section 3 we presented experiments using two of the most popular GNN architectures: GCN (Kipf and Welling, 2016) and GAT (Veličković et al., 2017). However, our theoretical results are not limited by the choice of GNN, so long as that GNN meets assumption A2 (which is light). Below we repeat the school experiment using two different choices of GNN: GraphSAGE (Hamilton et al., 2017), and JKNet (Xu et al., 2018).

Similar to the results in the main text, we see that coverage remains valid under these GNNs. Further, we see that the unfolded embeddings out-perform the block embeddings, both in terms of accuracy and prediction set size, in both the transductive and semi-inductive regimes.

| Embedding | School | |
|---|---|---|
| | Trans. | Semi-ind. |
| Block GraphSAGE | $0.856 \pm 0.014$ | $0.102 \pm 0.025$ |
| Unfolded GraphSAGE | $\mathbf{0.932 \pm 0.006}$ | $\mathbf{0.917 \pm 0.009}$ |
| Block JKNet | $0.885 \pm 0.009$ | $0.102 \pm 0.027$ |
| Unfolded JKNet | $\mathbf{0.930 \pm 0.007}$ | $\mathbf{0.909 \pm 0.009}$ |

(a) Accuracy. Bold values indicate higher accuracy.

| Embedding | School | |
|---|---|---|
| | Trans. | Semi-ind. |
| Block GraphSAGE | $\mathbf{0.900 \pm 0.012}$ | $\mathbf{0.869 \pm 0.072}$ |
| Unfolded GraphSAGE | $\mathbf{0.902 \pm 0.011}$ | $\mathbf{0.922 \pm 0.027}$ |
| Block JKNet | $\mathbf{0.901 \pm 0.012}$ | $\mathbf{0.897 \pm 0.028}$ |
| Unfolded JKNet | $\mathbf{0.901 \pm 0.012}$ | $\mathbf{0.907 \pm 0.027}$ |

(b) Coverage. Bold values indicate validity with target $\geq 0.90$.

| Embedding | School | |
|---|---|---|
| | Trans. | Semi-ind. |
| Block GraphSAGE | $5.350 \pm 0.230$ | $8.632 \pm 0.683$ |
| Unfolded GraphSAGE | $\mathbf{3.879 \pm 0.978}$ | $\mathbf{3.194 \pm 0.659}$ |
| Block JKNet | $4.724 \pm 0.332$ | $8.946 \pm 0.189$ |
| Unfolded JKNet | $\mathbf{2.566 \pm 0.292}$ | $\mathbf{2.472 \pm 0.477}$ |

(c) Average set size. Bold values indicate smaller set sizes.

Table 12: Repeated runs on the school dataset using GraphSAGE and JKNet as the choice of GNN.

