# OpenReview forum: "Valid Conformal Prediction for Dynamic GNNs"
_ICLR.cc/2025/Conference — ICLR 2025 Poster_

### Official Review · Reviewer_69Z2 · 2024-10-27

**Soundness:** 3
**Presentation:** 3
**Contribution:** 2
**Rating:** 5
**Confidence:** 3

**Summary:**

The paper studies uncertainty prediction for dynamic graphs. For the representation learning of dynamic graphs, the paper leverages the unfolded adjacency matrix as input to GNNs and for uncertainty prediction, the paper follows procedures of conformal prediction that constructs provably valid prediction sets.

**Strengths:**

1. The paper studies an important problem relating to uncertainty quantification for GNN prediction.

2. The use of unfolded adjacency for GNNs on dynamics graphs is natural and promising.

3. The experiments show promise for the proposed method.

**Weaknesses:**

1. The paper is poorly written with many concepts and notations not sufficiently explained. In addition, the structure of the paper needs to be improved. For example, (1) in line 161, it is not clear what it means for 'appropriate ordering of pairs', and what does m here represent? (It is better add some examples) (2) What is the intuition of using dilated unfolding in line 176? How does line 181-185 work? It would be clearer if the explicit update form is written under the example of GCN. (3) Algorithm 1 is introduced with no explanations on the steps. For example, what is a calibration set? (4) In theory, the key definitions, such as exchanebility and label equivariant are deferred to appendix, which is not ideal.

2. The developments are disconnected and it is thus not clear what are the key contributions of this work. The paper claims the contribution as a novel interface between conformal prediction and GNN. However, from the present version of the paper, it seems straightforward to combine the two to form conformal prediction on graphs. The consideration of dynamic graphs in this paper is novel but the use of unfolded adjacency has been considered previously for spectral embedding.

3. The scalability with the use of unfolded adj is poor.

**Questions:**

1. In A2, is label equivariant the same as permutation equivariant?

2. In section 2.1, can you formally prove the exchangebility of UGCN while BD GCN does not satisfy the exchangebiity?

3. Why Algorithm 1 is present in the main paper but Algorithm 2 is used for experiments?

---

> ### Author Response · Authors · 2024-11-18
>
> > The paper is poorly written with many concepts and notations not sufficiently explained. In addition, the structure of the paper needs to be improved.
>
> "Poorly written'' is really unfair. Other reviewers found our writing clear, the separation of formal mathematics and intuitive explanation helpful. In retrospect we may have accidentally made assumptions about background knowledge (particularly on conformal prediction, e.g. "calibration sets''). We will rectify these to make the paper more generally accessible.
>
> > (1) in line 161, it is not clear what it means for 'appropriate ordering of pairs', and what does m here represent? (It is better add some examples)
>
> This is a great suggestion, thanks. Working on this now, will update you when the revised PDF is ready.
>
> > (2) What is the intuition of using dilated unfolding in line 176? How does line 181-185 work? It would be clearer if the explicit update form is written under the example of GCN.
>
> It is surprising that this works, isn't it? The intuition, roughly, is to think of a dynamic graph as containing two families of nodes: global nodes, and time-localised nodes. $\mathbf{A}^{\text{UNF}}$ is a bipartite graph which connects them.  We do not understand what is meant by "the explicit update form''.
>
> > (3) Algorithm 1 is introduced with no explanations on the steps. For example, what is a calibration set?
>
> Calibration sets were introduced in the conformal literature and explained only via a formal definition and Figure 1, so we take the reviewer's point. We add on first introduction "The data are split into four sets: _training_ which is used to learn the GNN parameters; _validation_, which is out-of-sample performance evaluation for the GNN; _calibration_, which is used to train the conformal model, and _test_, for the final performance metric." and add a brief heurisitic description of this (standard) algorithm: "the algorithm proceeds as follows. First, we fit the GNN to the training data. Then, we compute the model predictions on the calibration set, and compare those to the observed labels using a non-conformity score. Finally, for the test pair, we return a confidence set which contains labels giving scores which aren't "too large'', relative to the calibration set. ''
>
> > (4) In theory, the key definitions, such as exchanebility and label equivariant are deferred to appendix, which is not ideal.
>
> Happy to move the definitions into the main text.
>
> > The developments are disconnected and it is thus not clear what are the key contributions of this work. The paper claims the contribution as a novel interface between conformal prediction and GNN. However, from the present version of the paper, it seems straightforward to combine the two to form conformal prediction on graphs. The consideration of dynamic graphs in this paper is novel but the use of unfolded adjacency has been considered previously for spectral embedding.
>
> The best ideas are often simple (Metropolis-Hasting, back-propagation, etc); simplicity is _never_ a good reason to reject. Instead, we would recommend that the reviewer decide whether the idea is _novel_, _useful_, and _non-obvious_. (See answer to wToC for additional details.)
>
> > The scalability with the use of unfolded adj is poor.
>
> The scaleability of unfolded GNN is the same as block GNN (line 483). There is a difference in _parallelisation_, but as we state (line 524) a pathway to massive-scale analysis is plausible for future work. A naive algorithm would be to analyse node-times for random subsets of nodes. A better algorithm would be to apply random-walk approaches to obtain neighbourhood sets. Such _algorithmic_ improvement is appropriate for future work and is not needed to prove that _in theory_ unfolding is an important tool for conformal dynamic GNN, and we have demonstrated utility _in practice_.
>
> > In A2, is label equivariant the same as permutation equivariant?
>
> Because this is a supervised context, label was a confusing word to use here, apologies. We will go with permutation equivariant.
>
> > In section 2.1, can you formally prove the exchangebility of UGCN while BD GCN does not satisfy the exchangebiity?
>
> We will provide references showing GCN is permutation equivariant. We will include a brief demonstration that even if A1 holds, the embedding $\hat{\mathbf{Y}}_{\mathbf{BD}}$ is not row-exchangeable. (We think this is overkill, given that it is already flagrantly shown in Figure 1)
>
> > Why Algorithm 1 is present in the main paper but Algorithm 2 is used for experiments?
>
> This is an LaTeX labelling error! Apologies. We apply Algorithm 1.

---

> > ### Comment · Reviewer_69Z2 · 2024-11-29
> > **Thank you for the response**
> >
> > I thank the authors for the detailed responses. Many of my concerns are addressed in the rebuttal. Thus I am increasing my score. However, I still think the idea in this paper is not significant and novel, which prevents me from increasing the score further.

---

> > > ### Author Response · Authors · 2024-12-03
> > >
> > > Thank you for improving the score.
> > >
> > > We must dispute the claim of non-novelty. Barring an enormous oversight on our part, we are the first to propose the use of unfoldings in the context of either: graph neural networks or conformal prediction.
> > >
> > > On significance: at present, outside of our paper, there is no way of obtaining valid prediction sets on dynamic graphs, and yet these represent the overwhelming majority of real-world graphs.

---

### Official Review · Reviewer_wToC · 2024-11-03

**Soundness:** 4
**Presentation:** 4
**Contribution:** 1
**Rating:** 3
**Confidence:** 4

**Summary:**

This paper focuses on dynamic graphs. More specifically, utilizing the tool of dilated unfolding, this paper expands the traditional GNN to dynamic graphs. No modification to GNN structure is required.

**Strengths:**

For a sequence of graphs, the paper cleverly utilizes dilated unfolding to make it a sparse matrix $A^{UNF}$. In the semi-inductive regime, the paper shows a proof guarantee of the algorithm. A comparison with the standard block GNN is provided in Sec 2.1 and clearly shows this advantage. Experiments in Sec 3.2 also support this.

**Weaknesses:**

I am not an expert in GNN, so please tell me if I am incorrect. It seems the innovation in this paper is incremental because dilated unfolding is known in Davis '23. The theoretical contributions (the proof in Sec. B.1 and B.2) also seem natural.

**Questions:**

Could you confirm your algorithm is by combining dilated unfolding (Davis '23) and a GNN? If that is the only contribution, could you please convince me there is enough innovation provided?

---

> ### Author Response · Authors · 2024-11-18
>
> > Could you confirm your algorithm is by combining dilated unfolding (Davis '23) and a GNN?
>
> Confirmed!
>
> > If that is the only contribution, could you please convince me there is enough innovation provided?
>
> The best ideas are often simple (Metropolis-Hasting, back-propagation, etc); simplicity is _never_ a good reason to reject. Instead, we would recommend that the reviewer decide whether the idea is _novel_, _useful_, and _non-obvious_.
>
> On novelty:
> It is nonsense to claim that because a certain graph representation has been proposed before, a new paper using it isn't novel. Shall we also reject any paper which uses an adjacency matrix?
>
> The dilated unfolding is used in Davis et al. 23 for unsupervised learning tasks such as clustering. Here we are doing supervised learning, such as classification. These are really very different domains, and Davis et al. do not allude to any possibilities (such as conformal prediction) in that direction. There is no overlap between the theory and models of both papers. In Davis et al. 23, an Erdos-Renyi model is used, i.e. $A_{ij}^{(t)} \sim \text{Bernoulli}(P^{(t)}_{ij})$ independently; here our model is more general, and can accommodate dependence between edges. Most of the work in Davis et al. 23 is to _establish_ exchangeability; here, most of the theoretical work is to establish _what sort of exchangeability_ we need (if any) for _conformal prediction_, a concept which has no meaning in unsupervised settings.
>
>
> On usefulness:
> 1. Obtaining prediction intervals for dynamic networks is a significant problem for society (see intro), completely unsolved until this paper.
> 2. The solution proposed, precisely because it is so simple, could reasonably be implemented on large GNNs already in deployment.
>
> On non-obvious:
> 1. (Theory) Demonstrating validity has required substantial and non-trivial novel analysis. Surely the reviewer does not claim that anyone could have predicted that the transductive and temporal transductive regime required no further assumptions, whereas the semi-inductive regime required A1 + A2?
> 2. (Experiments) We have conducted a comprehensive series of well-considered experiments, and provided the data and code, to uncover different interesting features of the algorithm in real data --- including smaller prediction sets than rival methods, prediction sets adapting to periods of uncertainty (see school data), and different failure modes.

---

> > ### Comment · Reviewer_wToC · 2024-11-24
> >
> > Dear authors
> >
> > I deeply appreciate your reply, which helps me better understand the paper. I am so sorry I cannot improve my rating.
> >
> > I am also deeply grateful for the AC's service. It is your effort to make such a nice conference!
> >
> > Best wishes
> > reviewer

---

### Official Review · Reviewer_X1HP · 2024-11-04

**Soundness:** 2
**Presentation:** 3
**Contribution:** 2
**Rating:** 5
**Confidence:** 2

**Summary:**

This paper works on applying conformal prediction on dynamic graphs. The key contribution of this paper is introducing a careful mathematical consideration of different inferences in modeling dynamic graphs. Valid conformal predictions are obtained in most of the scenarios and the authors provide real data examples to prove the theory.

**Strengths:**

It is interesting to use block GCN to prove the validity of applying conformal prediction on dynamic graphs. The authors provide detailed and sufficient explanations and analysis on different scenarios on dynamic graph tasks. Besides, the authors provide a variety of real data examples to show the effectiveness of their theory. The paper is well-written and organized.

**Weaknesses:**

This paper aims to apply conformal prediction to dynamic graphs. It should include more competitive baselines that use conformal predictions on graphs. And the backbone models should be more rather than simply using GCN and GAT.

**Questions:**

1. This question is similar to the concerns in the weakness. Do you have more experimental results on other backbones?
2. What is the current state-of-the-art algorithm applying conformal prediction on dynamic graphs or even static graphs? The authors should include more competitive baseline methods.

**Details Of Ethics Concerns:**

No ethic concerns

---

> ### Author Response · Authors · 2024-11-22
>
> >It should include more competitive baselines that use conformal predictions on graphs. And the backbone models should be more rather than simply using GCN and GAT.
>
> We would instead ask, what value would showing other backbone models add? The important point about unfolding is that it modifies the representation of the _data_ for _any GNN_ so that it will provide _valid predictions_ under well-defined assumptions (appropriate exchangeability).
>
> Whilst for any given dataset, it is possible that some clever block GNN will perform well empirically, _it can never be valid_ because the input data does not contain the information required - i.e. stability over time - for validity. Conformal inference is important not because of its empirical performance but because it offers a well-defined guarantee of validity, which is the focus of the paper.
>
> > This question is similar to the concerns in the weakness. Do you have more experimental results on other backbones?
>
> We do see how a demonstration of the above point could be better made through other backbone models, so we've added GraphSAGE [1] and JKNet [2], which have the same qualitative performance (See Appendix H in the updated manuscript). Others are plausible, if you have suggestions, though we note that many seem very slow (so far we've only had time to run the school example) and we see this as demonstrating the point that unfolding, and not the backbone model, is the key.
>
> [1] Hamilton, Ying and Leskovec 2017 "Inductive Representation Learning on Large Graphs", NeurIPS, https://cs.stanford.edu/people/jure/pubs/graphsage-nips17.pdf
>
> [2] Xu et al 2018, "Representation Learning on Graphs with Jumping Knowledge Networks" ICML, https://paperswithcode.com/paper/representation-learning-on-graphs-with
>
> > What is the current state-of-the-art algorithm applying conformal prediction on dynamic graphs or even static graphs? The authors should include more competitive baseline methods.
>
> There are no competing valid conformal prediction methods for dynamic graphs (in the sense we mean), which is why we view this contribution as important.
>
> There are valid approaches for static graphs, for which our literature review (line 48; line 64-80) contains the approaches we are aware of (and none are suitable for comparison in the dynamic graph case).

---

### Official Review · Reviewer_bSGQ · 2024-11-04

**Soundness:** 3
**Presentation:** 3
**Contribution:** 3
**Rating:** 8
**Confidence:** 3

**Summary:**

The authors established a novel conformal prediction method for dynamic GNNs, where edges evolve over time, like transport and social networks. The primary contributions are as follows:

1. Dynamic Graph Representation for GNNs: an "unfolding" technique for representations of dynamic graphs has been proposed to achieve reliable prediction sets, compared to the original block representations.

2. Inference Scenarios: The authors fully discussed various inference scenarios specific to dynamic graphs, including transductive, temporal transductive, and semi-inductive regimes, with valid and reasonable assumptions if needed.

3. Empirical Validation and Practical Applications: Using several real-world datasets as well as simulated data, the proposed method demonstrates improved metrics over baseline approaches, with particular advantages in the semi-inductive case, showing great potential in analyzing various dynamic graph systems.

**Strengths:**

1. Originality: One innovation in this paper is introducing the “unfolded” dynamic graph representation, which allows standard GNNs to process dynamic graphs while maintaining the validity of CP techniques. Additionally, the paper extends CP applications to multiple dynamic graph inference cases, for example, semi-inductive regimes, which is pretty challenging in discovering missing labels.

2. Quality and Clarity: The authors provided clean and rigorous theoretical analysis in explaining how its method achieves valid predictions under varied dynamic settings, supported by well-defined assumptions and conditions for applicability in different inference scenarios. Visualizations like bar graphs and tables further help readers understand the reasonableness and benefits of the unfolded approach compared to the baseline.

3. Significance: The paper addresses an important issue in uncertainty quantification for dynamic GNNs, where given reasonable prediction intervals over time is essential. The basic idea for this approach seems to have great potential for scaling up, therefore advancing the applicability of GNNs with evolving edges.

**Weaknesses:**

1. Semi-Inductive Settings:

As the authors acknowledged, the assumption of exchangeability may not hold in many real-world cases, necessary discussions for mitigating it would be appreciated. For instance, if adapting robust techniques from time-series, like basic ARIMAs, or even neural SDEs might be useful.

2. Understanding UQ:

Though I acknowledged the authors’ efforts in providing extensive metrics, it would be insightful to how to interpret the size of prediction sets correlates with meaningful uncertainty in a dynamic graph. This could help researchers from other backgrounds understand the logic, and make benefits for them. Making a plot showing the coverage (if possible) will make the paper more solid.

3. Miscellaneous:

I suggested the reviewers consider the following issues, and if time allows, do some elaboration.

a) If there is any other way to reshape the dynamic graph from $(T, N, N)$ into a 2D-shape, e.g. compared the performance the reshaped matrix in shape $(T, N^2)$.

b) Another simplification on the current $A^{UHF}$ might be: $A^{(1)}$ at top left, and the remaining unfolded $\mathcal{A}$ dilated in the same way. I would consider if this will somehow reduce the complexity, especially when $T << N$.

c) In this paper the dynamical graphs always mean edge-evolving ones. There are also cases where nodes are evolving with time, or even both edges and nodes are changing, e.g. pandemic networks (with different types of nodes, and some nodes will disappear permanently). It is appreciated if authors can do some brainstorming on this topic to provide meaningful improvements.

**Questions:**

Besides several concerns that mentioned in the weakness part, here are several questions regarding the paper details:

Line 88: why requires a bipartite graph? Needs to explain.

Lines 161-165: for better understanding, I would prefer the authors directly used $m$ over $m + 1$ (surely some other formula like
Algorithm 1 also needs to be changed). But this is more neat.

Line 165: it is not encouraged to mix superscripts and subscripts in notation, especially when they share the same meanings (e.g. test pairs and calibration pairs).

Line 179: do we have any differences if we use row- rather than column-concatenation? I guess they might be the same. Please think about this.

Line 222: does column swapping means permutation? For example, is it equivalent to say, there exists a permutation matrix $P$. such that $PA^{UNF}P^T=A^{UNF, swapped}$?

Section 2.1 and Figure 1: If you can plot $A^{(1)}$ as a 2D heatmap, or the original graph using tools like networkx package, it will be better for illustration rather than just representation scatter plots. For example, the density of nodes or the color of heatmap may clear show what happened in this toy system.

Figure 2: needs a short comment on SBM evolving edges. Why is that?

---

> ### Author Response · Authors · 2024-11-22
>
> > As the authors acknowledged, the assumption of exchangeability may not hold in many real-world cases, necessary discussions for mitigating it would be appreciated. For instance, if adapting robust techniques from time-series, like basic ARIMAs, or even neural SDEs might be useful.
>
> This is an interesting direction. In our discussion (lines 526-528) we briefly mentioned how works from Barber et al., 2023 and Clarkson, 2023 could be applied downstream of our UGNN embeddings to aid performance in settings where exchangeability may not hold. Barber et al. 2023 introduce a conformal framework in which the nonconformity scores of more "trusted" data points are assigned a larger weight when computing the prediction set.  Clarkson, 2023 then applied this work to GNNs, proposing a nonconformity score known as neighbourhood adaptive prediction sets (NAPS), where nodes closest to the test point were weighted more strongly.
>
> So in the case of our trade example, assigning higher weightings to more recent data points will likely aid the performance of our method. However, as drift still exists between the most recent labelled nodes and the test nodes, assumption A1 will still not hold, and so conformal prediction will not generally hold, even after applying this. We therefore view these techniques as critical to improve predictive performance for particular tasks, but out of scope of this paper which is about validity.
>
> > Though I acknowledged the authors’ efforts in providing extensive metrics, it would be insightful to how to interpret the size of prediction sets correlates with meaningful uncertainty in a dynamic graph. This could help researchers from other backgrounds understand the logic, and make benefits for them. Making a plot showing the coverage (if possible) will make the paper more solid.
>
> We were not completely sure what was being requested here, perhaps you can clarify? In Figure 4 we show the prediction sets over time for the school example, which demonstrates variation in when uncertainty is present under different conformal models.
>
> > a) Is there is any other way to reshape the dynamic graph from (T, N, N) into a 2D-shape, e.g. compare the performance of the reshaped matrix in shape (T, N^2)
>
> This is another very interesting question, being discussed in the unfolding literature in the context of spectral embedding; e.g. https://arxiv.org/html/2410.09810v1. There may be advantages for other shapes which make different assumptions about the underlying exchangability. We see this as a different direction to that of conformal inference for dynamic graphs, best explored in separate work.
>
> > b) Another simplification on the current A^UNF might be: A^1 at top left, and the remaining unfolded calA dilated in the same way. I would consider if this will somehow reduce the complexity, especially when T<<N.
>
> A critical aspect of unfolding is to maintain exchangeability, i.e. that reordering rows and columns appropriately does not affect the embedding. The structure we interpret from this suggestion favours A_1 as special and therefore not exchangeable. More generally, it is important to consider the motivation changing the representation. Using a sparse representation the dilation costs only a factor of 2 more compute, for which we gain symmetry (and hence applicability of many more algorithms) as well as theoretical benefits (as in our paper). Innovations in representation would be an interesting contribution that are distinct from our goal of bringing conformal inference to dynamic GNNs.
>
>
> > c) In this paper the dynamical graphs always mean edge-evolving ones. There are also cases where nodes are evolving with time, or even both edges and nodes are changing, e.g. pandemic networks (with different types of nodes, and some nodes will disappear permanently). It is appreciated if authors can do some brainstorming on this topic to provide meaningful improvements.
>
> This is a fascinating topic that would bear its own research paper by itself due to the many potential options. One is to simply unfold the node classes (e.g. infection status); another would be to turn to tensor representations discussed above - to which GNNs naturally generalise. The critical aspect is that unfolding leads to exchangeability properties only with respect to properties that are jointly anchored, i.e. share the same "rows" (and whatever else we anchor on, if we view e.g. infection status as a third dimension in a 3d array).
>
> [Response continues below...]

---

> > ### Author Response · Authors · 2024-11-22
> >
> > [Response continued]
> >
> > > Line 88: why requires a bipartite graph? Needs to explain.
> >
> > We do not _require_ A^(t) to be a bipartite graph; all we require is that each A^(t) has the same number of rows. As this assumption is light, this _allows_ A^(t) to take many forms, including directed, bipartite or weighted graphs - however, we do not have any requirements on the directedness, number of parties or weightedness.
> >
> > > Lines 161-165: for better understanding, I would prefer the authors directly used m
> >  over m+1 (surely some other formula like Algorithm 1 also needs to be changed). But this is more neat.
> >
> > Thank you for this suggestion. We have defined m so that we write m everywhere that m+1 previously appeared.
> >
> > > Line 165: it is not encouraged to mix superscripts and subscripts in notation, especially when they share the same meanings (e.g. test pairs and calibration pairs).
> >
> > We have updated the notation to consistently use subscripts. The superscripts referred to sets whilst the subscripts to elements; this whole section was rewritten to make it clearer.
> >
> > > Line 179: do we have any differences if we use row- rather than column-concatenation? I guess they might be the same. Please think about this.
> >
> > They are the same for symmetric matrices. If we row-concatenate, then we treat rows as dynamic and columns as "anchors", so swapping whether "links into" or "links from" are being embedded dynamically. (Column concatenation can be maintained by transposing each $A^{(t)}$ first.)
> >
> > > Line 222: does column swapping means permutation? For example, is it equivalent to say, there exists a permutation matrix P. such that P A^UNF P^T = A^UNF,swapped ?
> >
> > Yes, exactly!
> >
> >
> > > Section 2.1 and Figure 1: If you can plot A^(1) as a 2D heatmap, or the original graph using tools like networkx package, it will be better for illustration rather than just representation scatter plots. For example, the density of nodes or the color of heatmap may clear show what happened in this toy system.
> >
> > We are space-limited to show more in Fig 1. Is the point to show that Fig 1c have the same A^(t) distribution?
> >
> > If so, we have stated in Section 2.1 (lines 251 and 252 in the original submission) that our toy system draws A^(1) and A^(2) from the same distribution (i.e. "no change between time points", as stated in the figure). Therefore, we feel that displaying the heatmap of A^(1) and A^(2) will add little and add a complexity overhead for decoding (two plots for time, but two different plots for embeddings).
> >
> > > Figure 2: needs a short comment on SBM evolving edges. Why is that?
> >
> > The evolution was random. The expected number of edges in the SBM changes with the B^(t) matrix. A B^(t) matrix with larger values therefore results in a network with more edges. In this case, time point 4 (fewest edges) likely corresponds to the case where s_1=s_2=s_3=0.08, and time point 6 (most edges) likely corresponds to the case where s_1=s_2=s_3=0.16.
> >
> > We have added a comment explaining this when we introduce the SBM system in Section 3 ("The ordering of these time points is random, leading to dynamic total edge count (Figure 2).".

---

### Official Review · Reviewer_7kpc · 2024-11-07

**Soundness:** 3
**Presentation:** 3
**Contribution:** 3
**Rating:** 8
**Confidence:** 4

**Summary:**

The purpose of this paper is to provide valid prediction sets (with finite sample guarantees) achieved through conformal prediction for GNNs on dynamic graphs. In this setting, the graph evolves through time and can thus be represented as a set of adjacency matrices" nodes are fixed, but edges can appear and disappear at each time point. Each node is also endowed with a set of potentially varying features and labels.

To provide prediction sets (ie, sets of labels for each nodes such that the real label is within the set with, say, 95% probability), the authors suggest leveraging unfolding, a tool stemming from the tensor literature. The procedure they describe is as follows:
- 1- create an unfolded adjacency by concatenating all the adjacency matrices (columnwise),  yielding a matrix of size 2nT  * 2nT. Same for the features: X_unf is of size nTp.
- 2- apply a GNN on the unfolded adjacencies and features to output a set of global representations for each of the nodes (meaning, global across time points), and "local" representations (node/time pairs).
- 3 -compute conformity scores to output prediction sets.

The authors  show that this procedure can help them get uncertainty estimates for 3 types of scenarios: transductive, timewise transductive, and semi-transductive.
The authors then proceed to show the results of numerous experiments (synthetic and on real data). More specifically, they use as benchmark a block GNN (GNN fitted on the block diagonal adjacency, essentially treating each snapshot as a different graph). The authors show that the block approach creates "block" effects when there should be none, and this also results in higher accuracy of the UGNNs over the block approach, and lower set sizes.

**Strengths:**

This paper proposes an interesting approach to uncertainty quantification in the context of dynamic graph. The method that they propose is new, but leverages existing methods in the GNN and conformal prediction literature.

The examples chosen by the authors are quite strong and compelling.

**Weaknesses:**

On the whole, I think this is a good paper, but perhaps a couple of modifications would clarify certain aspects:
1- Some of the notations could be clarified. For instance, I found the part explaining the procedure a little confusing. More specifically:
- $\hat{X}^{UNF}$: since X is already used to describe features, another letter would be preferable here. I originally thought that it meant the unfolded features.
- Similarly,  $\hat{Y}^{UNF},$ I thought this meant the unfolded labels.
- It would be great to add the dimensions (only  $\hat{X}^{UNF}$ is indicated to have dimension $n_r \times d$).
- Would it be possible to replace $n_r$ by $n$? I personally find the subscript to be more justified to indicate a refinement, such as when, say, there would be different numbers of nodes across time points. I (personally) find that here, it invites more questions than would be necessary.

2- the UGNN framework could be further detailed:
- I am personally not familiar with the dilated unfolding approach of Davis, and consequently don't really understand the training procedure. $Y^{UNF}$ corresponds to the time-node pairs, so they should be trained with the node/pair labels?

3- the current challenges in deploying CP to graphs could be clearer: it seems to me that the main contribution of the paper is the unfolding mechanism, that allows (a) better representation of nodes in embedding spaces, and that consequently (b)  lends itself well to UQ using CP. Maybe a reformulation of the introduction highlighting that some of the challenges in CP on graph embeddings would be a distribution shift of the embeddings if there is a batch effect could help highlight the contribution of the method. Currently, it is succintly mentioned ("de-alignment between embeddings across time points"), but I think this should be expanded upon to highlight current challenges and set the context a bit more clearly.

**Questions:**

My questions pertain the implementation of the method (see weaknesses).

---

> ### Author Response · Authors · 2024-11-22
>
> >  1- Some of the notations could be clarified. For instance, I found the part explaining the procedure a little confusing.
>
> We have clarified the procedure in response to your and other reviewers comments, rewriting how training, validation, calibration and testing data are all used.
>
>
> > $X^{UNF}$: since X is already used to describe features, another letter would be preferable here. I originally thought that it $X^{UNF}$ meant the unfolded features. Similarly, I thought this $Y^{UNF}$ meant the unfolded labels.
>
> Good point; this was an artefact of combining standard notational practices from embedding and conformal prediction work. We have switched from X and Y to represent different network embeddings to U and V.
>
> > It would be great to add the dimensions (only $X^{UNF}$ is indicated to have dimension).
>
> Thanks - we have added dimensions.
>
> > Would it be possible to replace $n_r$ by $n$ ? I personally find the subscript to be more justified to indicate a refinement, such as when, say, there would be different numbers of nodes across time points.
>
> We have reorganised subscripts and superscripts, removing $n_r$ and replacing it with $p$.
>
> > I am personally not familiar with the dilated unfolding approach of Davis, and consequently don't really understand the training procedure. $Y^{UNF}$ corresponds to the time-node pairs, so they should be trained with the node/pair labels?
>
> Yes, this is correct. There was a typo in the definition of $\mathcal{G}$ which did not show the dependence on Y; We've updated the equation to make this clear.
>
> > 1- create an unfolded adjacency by concatenating all the adjacency matrices (columnwise), yielding a matrix of size 2nT * 2nT. Same for the features: X_unf is of size nTp.
>
>  The unfolded matrix we propose will is (n+nT) x (n+nT), as opposed to 2nT x 2nT.
>
> > 3- the current challenges in deploying CP to graphs could be clearer: it seems to me that the main contribution of the paper is the unfolding mechanism, that allows (a) better representation of nodes in embedding spaces, and that consequently (b) lends itself well to UQ using CP.
>
> This is a very good suggestion. To be clear, the principal challenge we are addressing is CP on _dynamic_ graphs. We have elaborated on the problem in the introduction. ``The unsolved problem we address ...''

---

> ### Comment · Reviewer_7kpc · 2024-11-28
> **RE: authors' answer**
>
> Thank you for the clarifications. I think the new notations considerably improve the paper's readability, and the procedure is much clearer to me. I appreciate as well the edits on the introduction.
>
>
> I am increasing my score to an 8: the contributions of the paper are substantial enough to warrant acceptance. The authors combine successfully two simple concepts (CP and unfolding), in a manner that works well (as shown by their experiments as well). I think this is a very nice addition to the budding GNN + CP literature. My only first reservation was due to (what I thought were) confusing notations, which have been improved by the authors.

---

### Author Response · Authors · 2024-11-22

We would like to thank the reviewers for their time and expertise. We appreciate the positive comments, in particular, regarding the novelty and significance of using an unfolding representation of dynamic networks to address the important issue of uncertainty quantification for dynamic GNNs. We are pleased that the reviewers have noted the simplicity of our technique, naturally combining existing ideas from dilated unfolding (itself inspired from spectral embeddings), and using the theoretical results about stability to provide a rigorous and justifiable setting for conformal prediction on GNNs for the first time. We strongly believe that the ideas outlined in this paper are _novel_, _useful_, and _non-obvious_, and represent a significant contribution to uncertainty quantification in dynamic networks. The answer to wToC contains an expanded discussion.

Several reviewers highlighted areas where our problem setup introduced unnecessary complications. We apologise and hope you can appreciate the difficulty of defining notation for two previously independent literatures. To address this we have completely rewritten the "Problem setup" and "Proposed approach", for which reviewer comments were very helpful. We hope that you will now find the problem statement much easier to understand, and if your review was negatively impacted by this, please consider revising in light of the revisions.

We have added the extra experiments requested on two further GNN architectures. As with GAT and GCN, these agree with theoretical expectations that current approaches (i.e. block GNNs) lack the information required to make stable predictions across time. Conversely, because unfolding provides a representation in which nodes that behave the same receive the same embedding (exchangeability), these behave properly for prediction tasks about new time periods (semi-inductive and temporal transductive regimes).

---

### Meta-Review · Area_Chair_aece · 2024-12-16

**Metareview:**

The paper proposes a method for achieving valid conformal prediction on dynamic graphs using graph neural networks. By leveraging an unfolded representation, the approach ensures valid prediction sets across various inference regimes.

Strengths: Novel integration of unfolding with conformal prediction for dynamic graphs. Strong theoretical guarantees under multiple inference regimes. Demonstrates practical relevance with compelling experimental results. Provides a simple and generalizable method adaptable to any GNN architecture.

Weaknesses: Initial unclear notation and explanations. Limited discussion on addressing exchangeability assumptions in real-world scenarios. Scalability challenges with large dynamic graphs. Lack of comparison with more diverse baseline methods.

**Additional Comments On Reviewer Discussion:**

The revisions addressed key concerns effectively.

---

### Decision · Program_Chairs · 2025-01-22

Accept (Poster)